# Is Pre-Training Truly Better than Meta-Learning?

## Abstract

In the context of few-shot learning, it is currently believed that a fixed pre-trained (PT) model, along with fine-tuning the final layer during evaluation, outperforms standard meta-learning algorithms. We re-evaluate these claims under an in-depth empirical examination of an extensive set of formally diverse datasets and compare PT to Model Agnostic Meta-Learning (MAML). Unlike previous work, we emphasize a fair comparison by using: the same architecture, and all models trained to convergence. Crucially, we use a more rigorous statistical tool – the effect size (Cohen's d) – to determine the practical significance of the difference between a model trained with PT vs. a MAML. We then use a previously proposed metric – the diversity coefficient – to compute the average formal diversity of a dataset. Using this analysis, we demonstrate the following: 1. when the formal diversity of a dataset is low, PT beats MAML on average and 2. when the formal diversity is high, MAML beats PT on average 3. yet overall, the difference in the two methods is zero on average using the effect size The caveat for the first two observations is that the magnitude of the average difference between PT vs. MAML using the effect size is low (according to classical statistical thresholds) – less than 0.2. Nevertheless, this observation is contrary to the currently held belief that a pre-trained model is always better than a meta-learning model and prove it by a counter example. Our extensive experiments consider 21 few-shot learning benchmarks, including the large-scale few-shot learning dataset Meta-Dataset and a NLP dataset. In addition, for our analysis, we trained 196 models to convergence. In addition, we provide a multimodal analysis and show the effect size difference is zero for GPT-2 PT and MAML models trained on Openwebtext. We, therefore, conclude that a pre-trained model does not always beat a meta-learned model and that the formal diversity of a dataset is a driving factor and when considering all datasets diversities there is no difference (as expected by the No Free Lunch Theorem).

## 1 Introduction

Rapid learning from a few examples is a hallmark of human intelligence. Inspired by the ambition to equip artificial agents with this capability, research in Artificial Intelligence (AI) has been significantly influenced by the domain of meta-learning, embodying the notion of "*learning to learn*" or "*learning to adapt*" (Andrychowicz et al., 2016). In addition, the broad success of Deep Learning in computer vision (Krizhevsky et al., 2012; He et al., 2015), natural language processing (Devlin et al., 2018; Brown et al., 2020), game playing (Silver et al., 2016; Mnih et al., 2013; Ye et al., 2021), theorem proving (Rabe et al.; Polu & Sutskever, 2020; Han et al.), code (Chen et al.) and more has lead Deep Learning to be routinely applied to the field of few-shot learning. Few-shot learning challenges a model's ability to quickly learn a new task from few examples and therefore has been a prominent research area for the application of meta-learning algorithms. However, recent findings (Tian et al., 2020) have demonstrated that a pre-trained model with a fixed embedding can surpass many sophisticated meta-learning algorithms in performance on a variety of few-shot learning benchmarks (Tian et al., 2020; Chen et al., 2019; 2020; Dhillon et al., 2019; Huang & Tao, 2019). It has been suggested that we need to "Re-think Few-Shot Image Classification; is a Good Embedding All You Need?". We challenge this view and re-evaluate meta-learning against pre-trained models – using Model Agnostic Meta-Learning (MAML) as the canonical meta-learning algorithm. Our novel re-evaluation uses three fundamental components: A. It employs a fair comparison amongst

pre-trained models vs MAML trained models by; using the same architecture, and all models trained to convergence. B. It uses a more rigorous statistical analysis technique – in particular, we use the effect size (Cohen's d) (Cohen, 1992) to characterize more carefully the difference in performance between the two methods. C. It takes into account the formal diversity of a dataset when conducting the statistical analysis.

All of this work is inspired by the key question: does explicitly training to "learn to learn" (i.e., meta-learn) improve the performance of a machine learning algorithm? We provide an answer in few-shot learning: in the low data diversity regime, pre-training is better on average than MAML, but in the high diversity regime, MAML is better. The caveat is that the magnitude of the effect size (Cohen's d) is less than 0.2 on average – which is considered small in statistics (Andrade, 2020; Lin et al., 2013; Cohen, 1992). However, when taken all dataset diversities into consideration, the effect size is close to zero (-0.00475), favoring neither algorithm – as would be expected from the No Free Lunch Theorem (NFLT) and previous work suggesting this (Miranda et al., 2022; Wolpert & Macready, 1997).

**Our key Contributions** are summarized as follows:

1. We clarify previous claims that pre-training is better than MAML through a fair, in-depth, and extensive study – using the effect size (Cohen's d). Therefore, we go beyond statistical significance to establish practical significance instead.

2. We do the analysis in a novel data-centric perspective by measuring the formal diversity of the datasets. This reveals an additional rich structure to our findings and consistency with the No Free Lunch Theorem (NFLT) (Wolpert & Macready, 1997).

3. We demonstrate:

    (a) When the formal diversity of a dataset is low, a Pre-trained model beats MAML on average (but with a small effect size)

    (b) When the formal diversity of a dataset is high, a MAML model beats a Pre-trained model on average (but with a small effect size).

    (c) When all dataset diversities are considered, neither method is better than the other (as expected by the NFLT).

## 2   Background

**Model-Agnostic Meta-Learning (MAML):** The MAML algorithm (Finn et al., 2017) is designed to meta-learn an optimal initialization of neural network parameters, thereby priming it for rapid gradient descent adaptations. This algorithm involves two core optimization loops: the outer loop, which primes the parameters for swift adaptation, and the inner loop, which executes the fast adaptation. During meta-testing/evaluation, the inner loop exclusively carries out the adaptation of the representation acquired from the outer loop.

**Pre-Training (PT) with a union of all the data:** Prior research (Tian et al., 2020) demonstrates that an initialization pre-trained with a union of all the data can supersede numerous meta-learning methods. Specifically, their methodology involves two phases: initially, they utilize a union of all labels in the few-shot learning benchmark during meta-training and undertake training with conventional pre-training (PT). Subsequently, during the meta-testing phase, they employ a standard inference method prevalent in transfer learning: extraction of a fixed feature from the neural network and a comprehensive fine-tuning of the final classification layer (the head) with LBGFS (Limited-memory Broyden–Fletcher–Goldfarb–Shannon algorithm).

**Effect Size (Cohen's d):** Cohen's d (Cohen, 1992) is a robust statistical tool designed to quantify the size or magnitude of an effect, irrespective of the sample size. This standardized measure of effect size allows for the comparison of results across different studies and domains. It is calculated by determining the difference between two means and dividing by the pooled standard deviation (approximately unbiased estimate of the combined standard deviations), providing a measure of effect size in terms of standard deviation units: $d = (\mu_1 - \mu_2)/pooled\_std(\sigma_1, \sigma_2)$. We explain the main reason for using this metric and its corresponding decision rule in section 3.

**Task2Vec Embeddings of tasks:** We use the Task2Vec diversity coefficient proposed in Miranda et al. (2022) to compute the formal diversity of a dataset (or a few-shot learning benchmark). To

understand the diversity coefficient, we explain how to compute Task2Vec (vectorial) embeddings of a task and briefly explain why it is a good vectorial embedding of a task. Task2Vec (Achille et al., 2019) embeds data (e.g., any batch) using the diagonal entries of the Fisher Information Matrix (FIM) using a fixed neural network (also called a **probe network**) after (partially) fine-tuning the final layer to solve the current task (or batch). Thus, the Task2Vec embedding of task $\tau$ is:

$$\vec{f}_{D_\tau, f_w} = Diag(\hat{F}_{D_\tau, f_w}) \tag{1}$$

where $Diag$ extracts the diagonal of the FIM:

$$\hat{F}_{D_\tau, f_w} = \mathbb{E}_{x,y} \nabla_w \log p(y \mid x, f_w) \nabla_w p(y \mid x, f_w)^\top$$

and $f_w$ is the fixed probe network with architecture $f$ with weights $w$, $x$ is sampled from the batch/data $D_\tau = \{(x_i, y_i)\}_{i=1}^n$ for task $\tau$, $y$ is sampled from the (empirical) posterior distribution using the probe network i.e., $p(y \mid x, f_w)$. This is a good embedding of tasks because the (diagonal) of the FIM indicates the most informative weights for solving the current task and thus serves as a unique fingerprint for task distribution. The Task2Vec authors (Achille et al., 2019) empirically validate their embeddings, e.g., Task2Vec embeddings cluster in a way that matches human semantic relations between different visual tasks (Achille et al., 2019) and Task2Vec yields (cosine) distance that positively correlation with taxonomical distances (Achille et al., 2019).

**Task2Vec Diversity coefficient:** The Task2Vec diversity coefficient is a formal quantitative metric proposed by Miranda et al. (2022) to approximate the effective number of tasks in a dataset. If the tasks are probability distributions, then this metric approximates the average distance between probability distributions. They validate it with synthetic experiments where the ground truth diversity is known. We further validate it in the supplementary section A.5. We show the intuitive notion that when different types of datasets are unioned to create a new dataset, the Task2Vec diversity coefficient increases. The *Task2Vec diversity coefficient* is defined as the expected (cosine) distance between Task2Vec embeddings of different tasks (or data batches) for a fixed probe network from a few-shot learning benchmark/dataset $B$:

$$\hat{div}(B) = \mathbb{E}_{\tau_1, \tau_2} \mathbb{E}_{D_1, D_2} d(\vec{f}_{D_1, f_w}, \vec{f}_{D_2, f_w}) \tag{2}$$

where $F_{D_1, f_w}$ is the Task2Vec embedding of the training data $D_\tau = \{(x_i, y_i)\}_{i=1}^n$ for task $\hat{p}(x, y \mid \tau)$ that uses the fixed probe network $f_w$ with architecture $f$ and weights $w$. Note that $\tau_1, \tau_2$ are tasks sampled from the (meta) distribution of tasks $\hat{p}(\tau \mid B)$ for the current benchmark $B$, and $d$ is the cosine distance. It is worth restating that in this framework, a task is defined as an n-way, k-shot few-shot learning task. Each task thereby includes $n$ classes, each of which is sampled with $k$ examples. Usually, this is split into a support (train) set for fast adaptation and a query (test) set for evaluation.

## 3 STATISTICAL METHODS

In this section, we explain the main statistical methodology we use to analyze our results using the effect size (Cohen's d) (Cohen, 1992; Andrade, 2020) and its rationale. We want to emphasize that the main tool we used was the **effect size** (Cohen, 1992) described in Section 2.

### 3.1 DECISION RULE

**Summary of decision rule:** To determine if there was a significant difference between PT vs MAML we computed the meta-test accuracy of the two methods and determined if the difference in terms of the effect size was larger than what is common in machine learning (we use a 1 % standardized threshold by dividing by the pooled standard deviation (Rodríguez et al., 2020; Li et al., 2021)). If it is larger, we determine there is a significant difference and report which method is better by looking at the sign of the effect size. Otherwise, if the difference was small according to this threshold, we reported there was no significant difference. All our tables include the raw effect size differences.

More precisely, first compute a list of (meta) test accuracies on a batch of few-shot learning tasks for MAML and PT denoted as: $accs_{pt}$ and $accs_{maml i}$, where $i$ denotes the number of inner steps in MAML during (meta) testing. Compute the effect size (ES) of the difference of these. Compute the threshold $\delta_{1\%}$ by dividing 1 % by the pooled standard deviation of the current test accuracies, i.e.

$$Decision(accs_{pt}, accs_{maml_i}) = \begin{cases} \text{H0 (no diff.)} & \text{if } ES(accs_{pt} - accs_{maml_i}) \in [-\delta_{1\%}, \delta_{1\%}] \\ \text{H1 (pt)} & \text{if } ES(accs_{pt} - accs_{maml_i}) > \delta_{1\%} \\ \text{H1 (maml)} & \text{if } ES(accs_{pt} - accs_{maml_i}) < -\delta_{1\%} \end{cases}$$

(3)

divide $0.01$ by $std\_pooled(accs_{pt} - accs_{mamli})$. Then the decision rule in equation 3 to determine if to reject the null hypothesis or not. Note, you can also look at the effect size sign to determine which method performed best. If the sign is positive, then PT outperformed MAML, otherwise MAML performed better. For most of our experiments, we noticed $\delta_{1\%} \approx 0.06$, therefore, an effect size difference with magnitude less than $0.06$ meant there wasn't sufficient evidence to reject the null hypothesis i.e. there wasn't a significant difference between the methods.

## 3.2 JUSTIFICATION FOR EFFECT SIZE AND DECISION RULE

The principal rationale behind our choice for using effect size lies in the sizable sample/batch size we employed, ranging from 300-500. When we used t-tests, we obtained p-values and confidence intervals equal to zero. Therefore, we cannot meaningfully use these statistical methods and over-reject the null hypothesis, a known issue in statistics (Lin et al., 2013). In other words, in studies with large sample sizes, even tiny, unimportant differences can be statistically significant. Reporting effect size prevents this type of misleading conclusion (Lin et al., 2013).

We also seek a data-centric perspective to our analysis. Therefore, it was paramount that our analysis was robust across datasets. Using effect size allows for this comparison because it's a standardized measure, so it's not influenced by sample size or the units of measurement.

Effect size provides information about the magnitude or strength of the difference or relationship between variables, going beyond the mere existence of an effect. It offers more informative insights than solely determining whether a difference is significant.

In the end, every statistical decision test will have an arbitrary value that needs to be chosen and justified. For t-tests, it's the p-value, commonly set to 0.05. For confidence intervals is the confidence level, commonly set to 95%. For effect size, the standard used values are 0.2, 0.5, 0.8 (Andrade, 2020). In this setting, we choose the difference to be the standardized 1 %. We choose this value because it is the common performance gained needed in machine learning conferences, these papers are some evidence Rodríguez et al. (2020); Li et al. (2021). However, note is arbitrary, and therefore it is important to report all absolute effect sizes and raw meta-test accuracies. We do this in our work, and the meta-test accuracies are in the supplementary section A.2. We finalize this section by reminding the reader all statistical tests have assumptions and are imperfect.

## 4 EXPERIMENTS

We compare the performance of a pre-trained (PT) model against a MAML model using the effect size and the decision procedure described in section 3. To provide further insights, we organize our experiments results according to the formal diversity of the datasets. In particular, we divide the results into datasets with formally low and high diversity. We set the division of low vs high at $0.146$ Task2Vec diversity, because that was approximately the average diversity. In addition, this division roughly divided the datasets in an intuitive way: most datasets that were a union of four or more datasets had a diversity higher than $0.146$, while the rest were below. The only exception was Omniglot, which had a high diversity, but also contains a vast amount of visual concepts (over 1000). We present our experiment details in Section 4.1, a summary of our results in Section 4.2, and refer the reader to sections A.3 and A.4 for detailed results and explanations of datasets.

## 4.1 HYPERPARAMETER DETAILS

**Few-shot learning details:** All experiment had a 5-way, 20 shot setting – i.e, 5 train shot and 15 eval shot, for training MAML. For more details refer to section A.11.

**Hyperparameter Details for Resnet12 on low diversity datasets for Pre-training and fo/ho-MAML:** We used the Resnet12 architecture provided by Tian et al. (2020). The Adam optimizer (Kingma & Ba, 2017) was utilized with a constant learning rate of 1e-3. No learning rate scheduler was used. Training was performed for 600,000 iterations for pre-training and 160,000 first-order MAML iterations, with a batch size of 256. The outer loop consisted of 130,000 MAML iterations. We used an inner learning rate of 0.1 and 5 inner steps. No weight decay was applied. Training was performed on a single NVIDIA PU with at most 48GB memory select by a HPC automatically. All experiments were trained to convergence (less than 0.01 loss) and took on average at most 1 week. All implementations were done in PyTorch (Paszke et al., 2019).

**Hyperparameter Details for 5CNN on High Diversity Datasets:** We utilized the 5CNN architecture proposed in Tian et al. (2020) with varying filter sizes. The Adam optimizer Kingma & Ba (2017) was used with a learning rate of 1e-3 without any learning rate decay. A batch size of 256 was used for both pre-training and MAML training. No weight decay was applied. For pre-training, we trained for 200,000 iterations. For first-order MAML, we trained for 100,000 iterations with an inner loop of 5 steps and an inner learning rate of 0.1. We annealed the learning rate with a cosine scheduler with scheduler freq 2000 with minimum learning rate 1e-5 (similar to MAML++). All models were trained to convergence, which took approximately 1 week on a single NVIDIA GPU with at least 48GB of memory allocated by the HPC scheduler. All implementations were done in PyTorch Paszke et al. (2019).

**Hyperparameter Details for ResNet12 on High Diversity Benchmarks:** We utilized the ResNet12 architecture from Tian et al. (2020) for our experiments. The Adam optimizer (Kingma & Ba, 2017) was used with a learning rate of 1e-3 without any learning rate decay. For pre-training, we trained for 1 million iterations with a batch size of 256. For first-order MAML (Finn et al., 2017), we trained for 300,000 iterations also with a batch size of 256. The MAML outer loop consisted of 5 inner update steps with an inner learning rate of 0.1. No weight decay was applied. We annealed the learning rate with a cosine scheduler with scheduler freq 2000 with minimum learning rate 1e-5 (similar to MAML++). All models were trained to convergence on a single NVIDIA GPU with at least 48GB of memory allocated by the cluster scheduler. Training took approximately 1-2 week to converge for both pre-training and MAML. All implementations were done in PyTorch (Paszke et al., 2019).

**Hyperparameter Details for ResNet50 on High Diversity Meta-Dataset Benchmarks:** We utilized the ResNet50 architecture from Tian et al. (2020) in our experiments on the high diversity meta-dataset benchmarks. The Adafactor optimizer (Shazeer & Stern, 2018) was used with default settings and no learning rate decay. We used Adafactor from Seqfair because it had a setting with no hyperparameter choices, the memory benefits that we needed given our compute and the evidence of previous work showing the training was 2.5-fold faster (Miranda et al., 2023). For pre-training, we trained for 300,000 iterations with a batch size of 256. For first-order MAML (Finn et al., 2017), we trained for 140,000 iterations also with a batch size of 256. The MAML outer loop consisted of 5 inner update steps with Adafactor's default inner learning rate. We used Adafactor default annealing scheduler in Seqfair. Due to computational constraints, we limited the number of random seeds to 4 – especially given that MDS combined 10 large scale vision datasets that includes ImageNet. Pre-training and MAML training took approximately 1 month each to converge on NVIDIA GPUs with 48GB memory allocated automatically by the cluster scheduler. All implementations were done in PyTorch (Paszke et al., 2019).

**Hyperparameter Details for mini-Gpt2 on openwebtext:** We derived our architecture from nanoGPT[1]. We set a block size of 32 and create a model with 4 layers and 4 heads per layer with an embedding size of 192 to have a smaller model that is easy for experimentation that we dub mini-gpt2. The Adafactor optimizer and scheduler (Shazeer & Stern, 2018) were used with default settings (that is, no hyperparameters were specified) and no learning rate decay. In addition due to the memory benefits that we needed given our compute and the evidence of previous work showing the training was 2.5-fold faster (Miranda et al., 2023). Training was performed till visual convergence of loss curves was reached for all experiments. For MAML training, we set inner loop learning rate as 1e-3 and performed 5 inner update steps. Training was performed in a distributed setting on 4 NVIDIA PU with at most 48GB memory select by a HPC automatically. All implementations were done in PyTorch (Paszke et al., 2019).

---

[1]https://github.com/karpathy/nanoGPT

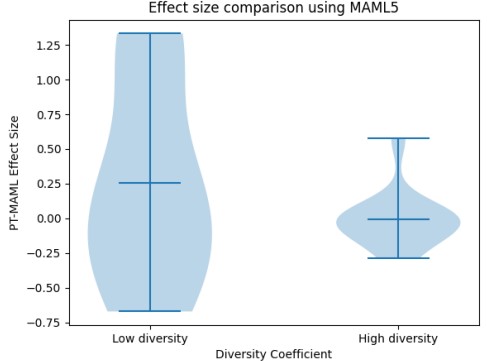 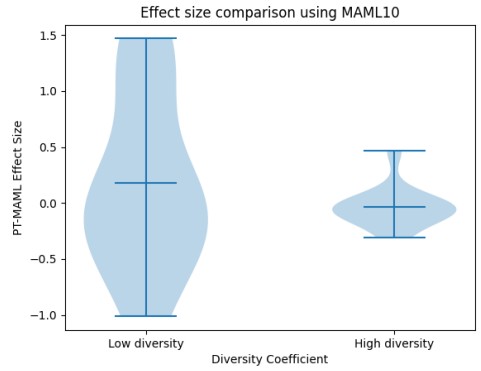

(a) Comparison of effect sizes running MAML5     (b) Comparison of effect sizes running MAML10

Figure 1: **Trend of observed effect sizes with diversity.** Here MAML5 and MAML10 represent running 5 inner adaptation steps and 10 inner adaptation steps respectively.

## 4.2 SUMMARY OF COMPARISON OF PRE-TRAINING (PT) VS MAML

We summarize the overall effect size for the four settings we studied. In the low diversity setting PT outperforms MAML but, in the high diversity setting MAML outperforms PT. In the low diversity setting, PT has a larger effect size relative to the effect size of MAML in the high diversity regime when analyzing the results in this way. In Figure 1 we show the trend of effect sizes observed for low and high diversity datasets using MAML5 and MAML10. In Table 2 we summarize all results and compare the low diversity vs the high diversity regime when there is a significant difference (i.e. H1). In more detail, when averaging all the low diversity results with an H1 (PT or MAML) decision, the overall effect size is 0.103, favoring PT. When averaging all the low diversity results, the overall effect size is -0.107 favoring MAML. However, note that this is considered low effect in classical statistics, as the regime for small, medium, high are roughly 0.2, 0.5, 0.8 (Andrade, 2020; Cohen, 1992).

Table 1: **Summary of experimental results when comparing a MAML model against a Pre-trained (PT) model using the average effect size (ES/Cohen's d) for each statistical decision rule.** We report the average effect sizes obtained for each decision and regime over all our experiments. Using the averages we can see that the difference in the low diversity regime is low but slightly favoring a pre-trained model, while in the high diversity a MAML model is favored. Note fo maml stands for first-order MAML and ho maml stands for higher order maml. Note NA stands for not applicable, since no experiment resulted in the decision procedure to choose that hypothesis.

| Setting | Decision: H0 | Decision: H1 (PT) | Decision: H1 (MAML) |
| --- | --- | --- | --- |
| Low diversity (fo maml) | 0.029 | 0.778 | -0.411 |
| Low diversity (ho maml) | NA | 0.669 | -0.717 |
| High diversity (all) | 0.0578 | 0.0638 | -0.148 |
| High diversity (5CNNs) | 0.00922 | 0.0795 | -0.192 |
| GPT2 (Openwebtext) | 0.00681 | NA | NA |

## 5 RELATED WORK

We suggest a data centric-oriented framework for examining meta-learning algorithms in response to the call to re-think meta-learning – especially in the context of few-shot learning (Tian et al., 2020). Also, the main distinction between our work and theirs (Tian et al., 2020) lies in: that we bring clarity and nuance to their surprising results by using more rigorous statistical analysis (through the effect size (Lin et al., 2013)), that we contextualize our finding by computing explicit data quality properties – like the formal diversity of a dataset, and that we use a much wider set of diverse datasets. In addition, unlike their work, ours focuses in ensuring a fair comparison by using a consistent neural

Table 2: **Summary of experimental results when comparing a MAML model against a Pre-trained (PT) model using the average effect size (ES/Cohen's d) for each statistical decision rule using all results at once.** The overall mean effect size for rejecting the null hypothesis (i.e. H1) was **0.103 for the low diversity setting** and **-0.109 for the high diversity setting.** This shows: 1. in the low diversity setting the H1 difference is low but marginally favoring a pre-trained model, while 2. in the high diversity the H1 difference is also low but marginally favoring a MAML model. Note fo maml stands for first-order MAML and ho maml stands for higher order MAML.

| Setting | Dec: H0 | Dec: H1 (PT) | Dec: H1 (MAML) |
|---|---|---|---|
| Low diversity (fo & ho maml) | 0.029 | 0.7292 | -0.556 |
| High diversity (all & 5CNNs & mini-GPT2) | 0.0521 | 0.0717 | -0.164 |

network architecture, optimizer, and all models trained to convergence. Another point of difference is their attainment of further accuracy gains through distillation, a method we have not analyzed but will consider for future work.

Some earlier work demonstrated that MAML operates chiefly through feature reuse (Raghu et al., 2020) rather than rapid learning, signifying that a model trained with MAML undergoes minimal alteration after the MAML adaptation. Our work diverges from theirs primarily in two ways: 1) we contrast MAML meta-trained models against models that are pre-trained with a union of all the data rather than solely comparing varying types of MAML models, and 2) we contextualize our analysis by explicitly analyzing properties of datasets, like formal diversity.

Related work also includes the predictability of adversarial transferability and transfer learning through extensive experimentation and a theoretical analysis (Liang et al., 2021). The primary difference between their work and ours is their primary focus on transfer learning, while we concentrate on meta-learning for few-shot learning. Additionally, we did not consider adversarial transferability, which forms a central part of their analysis.

Now we proceed to comment on datasets/benchmarks related to our work. The meta-dataset benchmark aims to create a larger and more diverse dataset for few-shot learning (Triantafillou et al., 2019). The key distinction between their work and ours is our use of a quantitative metric to measure the intrinsic diversity of a dataset, ad therefore bring more nuanced considerations beyond dataset size or even just class numbers. Another interesting work is the IBM Cross-Domain few-shot learning benchmark (Guo et al., 2019). They provide an interesting benchmark but we decided to leave it for future work since their setting is different from ours because cross-domain learning is out of scope for our work.

The work by Wang et al. (2021) proposes the concept of global labels, equivalent to what we call pre-training with all the data in this paper. Their theoretical analysis, however, is dependent on a fixed feature extractor, and fails to accommodate different feature extractors e.g. when comparing a Pre-trained model vs a MAML model. We address this in our empirical study.

## 6 DISCUSSION

Our paper presents an alternative explanation for claims that a Pre-trained model can often beat a model trained with meta-learning (Tian et al., 2020; Chen et al., 2019; 2020; Dhillon et al., 2019; Huang & Tao, 2019). In particular, we show that a more careful analysis – especially one that takes the formal dataset diversity into account and more rigorous statistical tools – can provide a nuanced truthful conclusion. Note, however, all statistical tools have assumptions and none are perfect – as the following discussion will exemplify.

For example, section 4.2 shows that when the overall average effect size across all high diversity datasets is used, we get a mean effect size of -0.105, favoring MAML. Similarly, across all low diversity datasets the effect size is 0.103, favoring PT. In addition, we get the very satisfying result that when all dataset diversities are considered, neither method is better – what would be expected according to the No Free Lunch Theorem (NFLT) (Wolpert & Macready, 1997). However, this effect size is considered low in classical statistics, since effect sizes of small, medium, high are roughly 0.2, 0.5, 0.8 (Cohen, 1992). However, to further nuance the analysis, if one uses the decision of the hypothesis test alone as in table 1, one might conclude that in the low diversity datasets Pre-training

is marginally better than MAML – since there was only one more time it outperformed MAML. A final nuance we'd like to add to our results is the variance of the effect size. We note the effect sizes ranges from 0.778 to -0.717. Meaning that when looking at individual experiments, the difference might be high for each individual run.

In addition, we want to emphasize that unlike previous work, we truly emphasized a fair comparison i.e. we used the same architecture, and all models trained to convergence. Previous results, especially Tian et al. (2020), compared different architectures and different meta-learning methods, thereby making it impossible to know the true source of improved performance. We disambiguate this and show that in datasets with low formal diversity, PT outperforms MAML, while MAML outperforms PT in datasets with high formal diversity.

We'd like to note that our low diversity results are in conflict with previous results in (Miranda et al., 2022). We conjecture the difference was mainly due to: 1. The batch size they used in conjunction with their use of Confidence Intervals (CIs) as their statistical analysis. Their experiments had (meta) batch size of 100, leading to overly large intersecting CIs that leads them to conclude pre-training and MAML are equivalent. 2. The number of models they trained is less than half of ours. We trained 40 models in total for the low diversity setting, and they did 18 (less than half of ours).

We'd also like to note that although MAML and Pre-training (PT) are marginally different (or that MAML is better than PT for high diversity datasets) – that MAML is still harder to train than PT. In particular, MAML requires an additional memory for the gradient in the forward pass and makes it harder to train for large models. Even though (Bronskill et al., 2021) memory efficient meta-learning might solve the memory issue, it is still less simple than pre-training.

In contrast to Tian et al. (2020), our work deliberately opts for a broad suite of datasets with varying formal diversities at the expense of exploring fewer meta-learning methods, which might be viewed as a potential limitation. This choice arises from Tian et al. (2020)'s implication that pre-training may outperform virtually all meta-learning algorithms (they considered). Our research, however, challenges this view (Tian et al., 2020), and demonstrates that even the simplest meta-learning algorithm, MAML, can outperform pre-training when the comparison employs fair and rigorous statistical analysis that considers the characteristics of the dataset like the formal diversity. In other words, we only need a single example (i.e., a single meta-learning algorithm) to provide a proof by counter example. In addition, by not considering as wide (and arguably unmotivated) of set of meta-learning algorithms we are able to conduct a more motivated A.7 investigation across a notably diverse array of datasets. Our approach underscores the importance of dataset properties on algorithm performance, thereby contributing a nuanced perspective to the ongoing discourse in the meta-learning domain.

The work by Kumar et al. (2022) provides an exploration of the effects of diversity in meta-learning. However, they focus mostly on sampling strategies, while we focused on the intrinsic diversity in the datasets/benchmarks themselves.

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

# A  SUPPLEMENTARY MATERIAL

## A.1  RELATED WORK (CONT.)

The ARC benchmark (Chollet, 2019) was designed with AGI in mind – potentially the ultimate meta-learner. However, ARC's focus is primarily on visual reasoning utilizing program synthesis techniques. We believe it's a promising path, but our work inspires extensions that transcend program synthesis approaches.

The study by Chen et al. (2021) offers, to our knowledge, the first non-vacuous generalization bounds for the (supervised) meta-learning setting. However, their results do not aim to differentiate classes of meta-learning, as our work attempts to do empirically.

The work by Wang et al. (2021) proposes the concept of global labels, equivalent to what we call USL in our paper. Their theoretical analysis, however, is dependent on a fixed feature extractor, and fails to accommodate different feature extractors that might be trained, such as comparing USL vs MAML directly in an end-to-end fashion. This was partially addressed theoretically and empirically in Miranda et al. (2022).

The study by Denevi et al. (2020) presents a theoretical treatment of meta-learning using meta-learners with closed-form equations derived from ridged regularization using fixed features. However, their results are highly theoretical, whereas ours focus on empirical results, and they do not explore their findings in the context of modern few-shot learning benchmarks like MiniImagenet, Cifar-fs, FC100, TieredImagenet, Meta-Dataset, etc. like we do.

The work by Goldblum et al. (2020) provides strong evidence that adaptation at test time is best done when the meta-trained model matches the adaptation it was meta-trained with. However, their results cannot beat Tian et al. (2020) and thus do not help separate the role of meta-training and pre-training (PT) with a union of all the data.

The study by Gao & Sener (2020) provides theoretical bounds of when the expected risk of MAML and DRS (Domain Randomized Search) by bounding the gradient norm. However, they do not provide in depth empirical analysis with respect to any real few-shot learning benchmarks like MiniImagenet or Cifar-fs.

The work by Rosenfeld et al. (2021) provides a theoretical analysis on the difference between interpolation and extrapolation in transfer learning. We believe this type of theory may be helpful as an inspiration to explore why in the high diversity regime there seems to be a difference between the performance of meta-learning and transfer learning or pre-trained methods.

Finally, the work by Miranda (2020b;a) first demonstrated that there exist synthetic datasets capable of exhibiting higher degrees of adaptation compared to the original work by Raghu et al. (2020). Their main focus was on comparing adapted MAML models vs. unadapted MAML models, a difference from our approach in this paper.

Previous work demonstrated that in datasets with low diversity, the difference MAML and pre-training is small (Miranda et al., 2022). While we substantiate these results to a degree, we introduce a nuance that, when evaluated through the statistical lens of effect size, pre-training can outperform MAML. This subtle detail underscores the critical role of the selected statistical measure in the comparative analysis of these algorithms. Finally, we provide the final piece of evidence to complete their story (Miranda et al., 2022), for high diversity datasets. Crucially, we include the large scale meta-dataset (mds) and demonstrate that merging/unioning datasets is an effective mechanism for increasing the formal diversity of a dataset.

BiT (Kolesnikov et al., 2020) is a study that demonstrates good performance on a wide set of datasets (20) using transfer learning by pre-training on a large (JFT-300M) scale vision dataset. They fine-tune the entire network (with SGD and momentum) during adaptation and provide heuristics for choosing the hyperparameters with the HyperRule heuristic. The main contrast with our work is that they do not do a direct fair comparison with meta-learning (like MAML) as we did. Contrary to this previous work that leaves the comparative merits of pre-training and meta-learning algorithms indeterminate, our work directly addresses this comparison as its primary focus. We think the authors Kolesnikov et al. (2020) should have compared their test time adaptation method that fine-tunes the entire paper to the one proposed by Tian et al. (2020) that uses optimal convergence at the final layer with

gradient-based BFGS fine-tuning. We note they use a large dataset for pre-training, and it's important to use such a dataset for the training of MAML to be able to do a fair comparison between MAML and pre-training. Our experiments on meta-dataset suggest that on large-scale formally diverse dataset MAML might be marginally better than pre-training.

Memory efficient meta-learning with large images (Bronskill et al., 2021) demonstrates that if one subsamples the support set (using their method called LIME) to meta-train many meta-learning algorithms, then one can match the performance of a pre-training network that has been fine-tuned with 50 steps. The main contrast between their work and ours is: 1. They use confidence intervals to separate pre-training methods vs meta-learning methods, while we use effect size 2. We add another level of structure to the analysis by separating the results in datasets that have a formal low diversity vs a formal high diversity. This analysis shows that MAML in fact can outperform pre-training, although being small when using the effect size as the measuring metric. We posit that our work, in conjunction with Bronskill et al. (2021), provides a complete perspective on meta-learning – where we conjecture that meta-learning methods in general marginally outperform pre-training methods. Their work (Bronskill et al., 2021) supports our counter-narrative that pre-training methods are always better.

The Vendi Score is a recently proposed formal diversity score different from the diversity coefficient proposed in Miranda et al. (2022). The Vendi score is mainly a more sophisticated aggregation method than an expectation given pair-wise comparisons. Their aggregation score is interesting, but it is unclear what the advantages of it are compared to a simpler expectation. For a sample of $n$ already embedded tasks (or data points), the Vendi score takes $O(n^3)$ (due to the use of eigenvalue computation), while ours uses the faster to compute expectation, which takes $O(n^2 - n/2) = O(n^2)$. We hope to explore the Vendi score in future work and compare it with the expectation aggregation score. However, the main weakness of the Vendi score that previous work address (Miranda et al., 2022) is the use of Task2Vec (Achille et al., 2019) to compute embeddings of tasks. The Vendi score assumes one already has such a comparison by assuming a Kernel/Grahm Matrix and unfortunately circumvents arguably the hardest part of the problem – computing effective embeddings a task. Their formulation also implies their analysis is mostly focused on individual data point diversity, while the diversity coefficient also works embedding tasks, batches, and even entire datasets.

The ranges of 0.2, 0.5, and 0.8 as small, medium and large effect sizes were proposed in Andrade (2020).

## A.2 ALL META-TEST ACCURACY OF A PRE-TRAINED (PT) MODEL VS. MAML

In this section, we report the raw meta-test accuracy of used (to compute the effect size) when comparing PT vs MAML models in the main body of the text section 4.

### A.2.1 META-TEST ACCURACY OF A PRE-TRAINED (PT) MODEL VS. A FIRST-ORDER (FO) MAML MODEL ON LOW DIVERSITY DATASETS

Meta-test accuracy (with 95% confidence intervals) of a Pre-trained (PT) model vs. a first-order (fo) MAML model on low diversity datasets are in table 3.

### A.2.2 META-TEST ACCURACY OF A PRE-TRAINED (PT) MODEL VS. A HIGHER-ORDER (HO) MAML MODEL ON LOW DIVERSITY DATASETS

Meta-test accuracy (with 95% confidence intervals) of a Pre-trained (PT) model vs. a higher-order (ho) MAML model on low diversity datasets are in table 4.

### A.2.3 META-TEST ACCURACY OF A PRE-TRAINED (PT) MODEL VS. A MAML MODEL ON HIGH DIVERSITY DATASETS

Meta-test accuracy (with 95% confidence intervals) of a pre-trained (PT) model vs. MAML model on high diversity datasets are in table 5.

Table 3: **Meta-Test accuracy of a Pre-trained (PT) model vs. a first-order (fo) MAML model with 95% confidence intervals on low diversity few-shot learning vision datasets.** We used a meta-batch size of 300 few-shot learning tasks. The datasets' diversity is low, as shown in table 11. Resnet12 has 1,427,525 parameters.

| Model (Dataset) | PT (test acc.) | MAML5 (test acc.) | MAML10 (test acc.) |
|---|---|---|---|
| Resnet12 (cifar-fs) | $0.755 \pm 0.0102$ | $0.779 \pm 0.00975$ | $0.786 \pm 0.00996$ |
| Resnet12 (fc100) | $0.438 \pm 0.00949$ | $0.458 \pm 0.00931$ | $0.459 \pm 0.00988$ |
| Resnet12 (mini-imagenet) | $0.719 \pm 0.00893$ | $0.685 \pm 0.00947$ | $0.706 \pm 0.0104$ |
| Resnet12 (tiered-imagenet) | $0.788 \pm 0.00945$ | $0.769 \pm 0.0107$ | $0.786 \pm 0.0107$ |
| Resnet12 (aircraft) | $0.592 \pm 0.010$ | $0.659 \pm 0.013$ | $0.685 \pm 0.011$ |
| Resnet12 (flower) | $0.928 \pm 0.005$ | $0.856 \pm 0.008$ | $0.870 \pm 0.007$ |
| Resnet12 (dtd) | $0.610 \pm 0.011$ | $0.511 \pm 0.012$ | $0.528 \pm 0.011$ |
| Resnet12 (delaunay) | $0.735 \pm 0.010$ | $0.614 \pm 0.012$ | $0.632 \pm 0.010$ |
| Resnet12 (cubirds) | $0.787 \pm 0.008$ | $0.829 \pm 0.008$ | $0.821 \pm 0.009$ |
| ResNet12 (vggair) | $0.727 \pm 0.027$ | $0.745 \pm 0.019$ | $0.760 \pm 0.019$ |
| ResNet12 (vggdtd) | $0.737 \pm 0.019$ | $0.701 \pm 0.022$ | $0.701 \pm 0.021$ |

Table 4: **Meta-Test accuracy of a Pre-trained (PT) model vs. a higher-order (ho) MAML model with 95% confidence intervals on low diversity few-shot learning vision datasets.** We used a meta-batch size of 300 few-shot learning tasks. The datasets' diversity is low, as shown in table 11. Resnet12 has 1,427,525 parameters.

| Model (Dataset) | PT (test acc.) | MAML5 (test acc.) | MAML10 (test acc.) |
|---|---|---|---|
| Resnet12 (cifar-fs) | $0.753 \pm 0.00941$ | $0.804 \pm 0.00982$ | $0.809 \pm 0.0107$ |
| Resnet12 (fc100) | $0.432 \pm 0.0102$ | $0.503 \pm 0.0100$ | $0.489 \pm 0.00988$ |
| Resnet12 (mini-imagenet) | $0.721 \pm 0.00889$ | $0.704 \pm 0.0100$ | $0.732 \pm 0.00952$ |
| Resnet12 (tiered-imagenet) | $0.791 \pm 0.00922$ | $0.771 \pm 0.0103$ | $0.695 \pm 0.0178$ |
| Resnet12 (aircraft) | $0.576 \pm 0.0116$ | $0.647 \pm 0.0127$ | $0.667 \pm 0.0112$ |
| Resnet12 (flower) | $0.921 \pm 0.00534$ | $0.902 \pm 0.00597$ | $0.899 \pm 0.00581$ |
| Resnet12 (dtd) | $0.600 \pm 0.0156$ | $0.501 \pm 0.0162$ | $0.519 \pm 0.0159$ |
| Resnet12 (delaunay) | $0.734 \pm 0.00984$ | $0.655 \pm 0.00981$ | $0.665 \pm 0.00986$ |
| Resnet12 (cubirds) | $0.785 \pm 0.00839$ | $0.857 \pm 0.00721$ | $0.857 \pm 0.00726$ |

### A.2.4 META-TEST ACCURACY OF A PRE-TRAINED (PT) MODEL VS. A MAML MODEL ON A VARYING SIZE OF 5CNN ON THE MICOD HIGH DIVERSITY DATASET

Meta-test accuracy (with 95% confidence intervals) of a Pre-trained (PT) model vs. MAML model on varying size of 5CNNs on the MICOD high diversity dataset are in table 6.

Table 5: **Meta-Test accuracy of a Pre-trained (PT) model vs. a MAML model with 95% confidence intervals on high diversity few-shot learning vision datasets.** Their diversity is high, as shown in table 12. Resnet12 has 1,427,525 parameters, while Resnet50 has 50,685,637 parameters.

| Model (Seeds) (Dataset) | PT (test acc.) | MAML5 (test acc.) | MAML10 (test acc.) |
|---|---|---|---|
| Resnet12 (fo maml) (omniglot) | $0.993 \pm 0.00148$ | $0.993 \pm 0.00139$ | $0.992 \pm 0.00164$ |
| Resnet12 (ho maml) (omniglot) | $0.994 \pm 0.00110$ | $0.985 \pm 0.00219$ | $0.988 \pm 0.00180$ |
| ResNet12 (fo maml) (mio) | $0.845 \pm 0.0121$ | $0.849 \pm 0.0136$ | $0.848 \pm 0.0133$ |
| ResNet12 (micod) | $0.778 \pm 0.0124$ | $0.781 \pm 0.0124$ | $0.786 \pm 0.0119$ |
| ResNet12 (hdb6-afdo) | $0.786 \pm 0.0205$ | $0.802 \pm 0.0190$ | $0.782 \pm 0.0178$ |
| ResNet12 (hdb7-afto) | $0.756 \pm 0.0227$ | $0.745 \pm 0.0226$ | $0.779 \pm 0.0216$ |
| ResNet12 (hdb8-cado) | $0.711 \pm 0.0218$ | $0.744 \pm 0.0227$ | $0.733 \pm 0.0208$ |
| ResNet12 (hdb9-cavdo) | $0.772 \pm 0.0210$ | $0.771 \pm 0.0207$ | $0.762 \pm 0.0211$ |
| ResNet12 (hdb10-micova) | $0.713 \pm 0.0244$ | $0.764 \pm 0.0177$ | $0.766 \pm 0.0167$ |
| Resnet50 (seed1 vs seed1) (mds) | $0.775 \pm 0.0133$ | $0.762 \pm 0.0133$ | $0.768 \pm 0.0144$ |
| Resnet50 (seed1 vs seed2) (mds) | $0.752 \pm 0.0138$ | $0.758 \pm 0.0150$ | $0.768 \pm 0.0144$ |
| Resnet50 (seed2 vs seed1) (mds) | $0.750 \pm 0.0141$ | $0.759 \pm 0.0151$ | $0.772 \pm 0.0152$ |
| Resnet50 (seed2 vs seed1) (mds) | $0.765 \pm 0.0135$ | $0.762 \pm 0.0147$ | $0.776 \pm 0.0143$ |

Table 6: **Meta-Test accuracy of a Pre-trained (PT) model vs. a MAML model with 95% confidence intervals on the high diversity MICOD few-shot learning vision dataset using varying size of 5CNNs.** We used a meta-batch size of 500 few-shot learning tasks. The Diversity Coefficient for the MICOD dataset is 0.174; details can be found in table 12.

| Filter Size (Dataset) | PT (test acc.) | MAML5 (test acc.) | MAML10 (test acc.) |
|---|---|---|---|
| 2 (micod) | $0.481 \pm 0.0205$ | $0.493 \pm 0.0197$ | $0.467 \pm 0.0184$ |
| 6 (micod) | $0.588 \pm 0.0169$ | $0.626 \pm 0.0189$ | $0.608 \pm 0.0178$ |
| 8 (micod) | $0.606 \pm 0.0161$ | $0.591 \pm 0.0178$ | $0.607 \pm 0.0184$ |
| 16 (micod) | $0.655 \pm 0.0149$ | $0.678 \pm 0.0154$ | $0.681 \pm 0.0157$ |
| 32 (micod) | $0.689 \pm 0.0151$ | $0.682 \pm 0.0150$ | $0.701 \pm 0.0154$ |
| 64 (micod) | $0.694 \pm 0.0135$ | $0.704 \pm 0.0155$ | $0.718 \pm 0.0152$ |
| 256 (micod) | $0.711 \pm 0.0139$ | $0.702 \pm 0.0163$ | $0.695 \pm 0.0156$ |
| 512 (micod) | $0.653 \pm 0.0175$ | $0.718 \pm 0.0158$ | $0.724 \pm 0.0154$ |

## A.3 ALL EFFECT SIZE RESULTS

We present the values of effect sizes obtained using different datasets and models under different regimes in the following sections. We note that we denote first order MAML as fo-maml and higher order MAML as ho-maml in the tables for brevity. We additionally denote using 5 internal adaptation steps with MAML by MAML5 and using 10 internal adaptation steps with MAML by MAML10.

### A.3.1 COMPARISON OF A PRE-TRAINED (PT) MODEL VS A FIRST-ORDER (FO) MAML MODEL ON LOW DIVERSITY DATASETS

Table 7 shows our experimental results for the low diversity setting when comparing a pre-trained (PT) model vs a first-order (fo) MAML model.

Overall, the effect size for H0 (no diff.) is 0.029, for H1 (PT) is 0.778, and for H1 (MAML) is -0.411. However, when averaging all low diversity results (as done in section 4.2) where there was a difference (H1 pt or maml), the overall effect size is 0.103, favoring PT.

### A.3.2 COMPARISON OF A PRE-TRAINED MODEL VS. A HIGHER-ORDER (HO) MAML MODEL ON LOW DIVERSITY DATASETS

Table 8 shows our experimental results for the low diversity setting when comparing a pre-trained (PT) model vs a higher-order (ho) MAML model.

Overall, the effect size for H0 (no diff.) is N/A (for PT vs ho-MAML H0 was never decided), for H1 (PT) is 0.669, and for H1 (MAML) is -0.717. However, when averaging all low diversity results (as

Table 7: **Meta-Test accuracy difference between a Pre-trained (PT) model vs. first-order (fo) MAML model using Effect Size (ES/Cohen's d) on low diversity few-shot learning vision datasets.** The diversity for them is low, as shown in table 11. Resnet12 has 1,427,525 parameters. We used a meta-batch size of 300 few-shot learning tasks.

| Model (Dataset) | PT-MAML5 (Decision) | PT-MAML10 (Decision) |
|---|---|---|
| Resnet12 (cifar-fs) | -0.266 (H1 maml5) | -0.342 (H1 maml10) |
| Resnet12 (fc100) | -0.251 (H1 maml5) | -0.248 (H1 maml10) |
| Resnet12 (mini-imagenet) | 0.413 (H1 pt) | 0.149 (H1 pt) |
| Resnet12 (tiered-imagenet) | 0.218 (H1 pt) | 0.0290 (H0 no diff.) |
| Resnet12 (aircraft) | -0.671 (H1 maml5) | -1.014 (H1 maml10) |
| Resnet12 (flower) | 1.224 (H1 pt) | 1.125 (H1 pt) |
| Resnet12 (dtd) | 1.332 (H1 pt) | 1.147 (H1 pt) |
| Resnet12 (delaunay) | 1.290 (H1 pt) | 1.262 (H1 pt) |
| Resnet12 (cubirds) | -0.572 (H1 maml5) | -0.452 (H1 maml10) |
| ResNet12 (vggair) | -0.105 (H1 maml5) | -0.195 (H1 maml10) |
| ResNet12 (vggdtd) | 0.200 (H1 pt) | 0.203 (H1 pt) |

done in section 4.2) where there was a difference (H1 pt or maml), the overall effect size is 0.103, favoring PT.

Table 8: **Meta-Test accuracy difference between a Pre-trained (PT) model vs higher-order (ho) MAML model using Effect Size (ES/Cohen's d) on low diversity few-shot learning vision datasets.** The diversity for them is low, as shown in table 11. Resnet12 has 1,427,525 parameters. We used a meta-batch size of 300 few-shot learning tasks.

| Model (Dataset) | PT-MAML5 (Decision) | PT-MAML10 (Decision) |
|---|---|---|
| Resnet12 (cifar-fs) | -0.602 (H1 maml5) | -0.628 (H1 maml10) |
| Resnet12 (fc100) | -0.800 (H1 maml5) | -0.643 (H1 maml10) |
| Resnet12 (mini-imagenet) | 0.205 (H1 pt) | -0.126 (H1 maml10) |
| Resnet12 (tiered-imagenet) | 0.236 (H1 pt) | 0.768 (H1 pt) |
| Resnet12 (aircraft) | -0.667 (H1 maml5) | -0.908 (H1 maml10) |
| Resnet12 (flower) | 0.382 (H1 pt) | 0.465 (H1 pt) |
| Resnet12 (dtd) | 1.240 (H1 pt) | 1.020 (H1 pt) |
| Resnet12 (delaunay) | 0.912 (H1 pt) | 0.793 (H1 pt) |
| Resnet12 (cubirds) | -1.043 (H1 maml5) | -1.044 (H1 maml10) |

### A.3.3 COMPARISON OF A PRE-TRAINED MODEL VS. A MAML MODEL ON HIGH DIVERSITY DATASETS

Table 9 shows our experimental results for the high diversity setting when comparing a pre-trained (PT) model vs a higher-order MAML model. For several rows we report averaged effect sizes over multiple seeds. When running pt for $p$ seeds and maml for $m$ seeds we report averages over all possible $p * m$ comparisons.

Overall, the effect size for H0 (no diff.) is 0.0721, for H1 (PT) is 0.0638, and for H1 (MAML) is -0.155. However, when averaging all high diversity results (as done in section 4.2) where there was a difference (H1 pt or maml), the overall effect size is -0.107, favoring MAML.

### A.3.4 COMPARISON OF A PRE-TRAINED (PT) MODEL VS A MAML MODEL ON A VARYING SIZE OF 5CNN ON HIGH DIVERSITY DATASET

Table 10 shows our experimental results for the high diversity setting when comparing a pre-trained (PT) model vs a MAML model using a varying size 5CNN.

Overall, the effect size for H0 (no diff.) is 0.00922, for H1 (PT) is 0.0795, and for H1 (MAML) is -0.192. However, when averaging all high diversity results (as done in section 4.2) where there was a difference (H1 pt or maml), the overall effect size is -0.107, favoring MAML.

Table 9: **Meta-Test accuracy difference between a Pre-trained (PT) model vs MAML model using Effect Size (ES/Cohen's d) on high diversity few-shot learning vision datasets as well as webtext using mini-Gpt2.** The diversity for them is high, as shown in table 12. Resnet12 has 1,427,525 parameters, while Resnet50 has 50,685,637 parameters. Note, mds stands for Meta-DataSet.

| Model (Seeds) (Dataset) | PT-MAML5 (Decision) | PT-MAML10 (Decision) |
|---|---|---|
| Resnet12 (fo-maml) (omniglot) | 0.00702 (H0 no diff.) | 0.0679 (H0 no diff.) |
| Resnet12 (ho-maml) (omniglot) | 0.577 (H0 no diff.) | 0.468 (H0 no diff.) |
| ResNet12 (fo-maml) (mio) | -0.0197 (H0 no diff.) | -0.0161 (H0 no diff.) |
| ResNet12 (seed1) (hdb4-micod) | -0.0166 (H0 no diff.) | -0.0559 (H0 no diff.) |
| ResNet12 (seed1) (hdb6-afdo) | -0.0919 (H1 maml5) | 0.0242 (H0 no diff.) |
| ResNet12 (seed1) (hdb7-afto) | 0.0528 (H1 pt) | -0.121 (H1 maml10) |
| ResNet12 (seed1) (hdb8-cado) | -0.167 (H1 maml5) | -0.116 (H1 maml10) |
| ResNet12 (seed1) (hdb9-cavdo) | 0.00798 (H0 no diff.) | 0.0552 (H1 pt) |
| ResNet12 (seed1) (hdb10-micova) | -0.287 (H1 maml5) | -0.308 (H1 maml10) |
| Mini-Gpt2 (30 seeds) (webtext) | 0.00681 (H0 no diff.) | -0.0114 (H0 no diff.) |
| ResNet12 (9 seeds) (hdb12) | -0.145 (H1 maml) | -0.190 (H1 maml) |
| ResNet12 (9 seeds) (hdb13) | -0.02955 (H0 no diff.) | -0.0345 (H0 no diff.) |
| ResNet12 (9 seeds) (hdb14) | 0.0371 (H0 no diff.) | -0.05745 (H0 no diff.) |
| Resnet50 (4 seeds) (mds) | 0.001375 (H0 no diff.) | -0.064275 (H1 maml10) |

Table 10: **Meta-Test accuracy difference between a Pre-trained (PT) model vs MAML model using Effect Size (ES/Cohen's d) on the high diversity MICOD few-shot learning vision dataset.** The Diversity Coefficient for the MICOD dataset was high with Task2Vec diversity coefficient of 0.174 as reported on table 12.

| Filter Size (Dataset) | PT-MAML5 (Decision) | PT-MAML10 (Decision) |
|---|---|---|
| 2 (micod) | -0.0533 (H1 maml) | 0.0624 (H1 pt) |
| 6 (micod) | -0.184 (H1 maml) | -0.100 (H1 maml) |
| 8 (micod) | 0.0794 (H1 pt) | -0.00121 (H0 no diff) |
| 16 (micod) | -0.131 (H1 maml) | -0.149 (H1 maml) |
| 32 (micod) | 0.0401 (H0 no diff) | -0.0689 (H1 maml) |
| 64 (micod) | -0.0588 (H0 no diff) | -0.145 (H1 maml) |
| 256 (micod) | 0.0568 (H0 no diff) | 0.0969 (H1 pt) |
| 512 (micod) | -0.341 (H1 maml) | -0.376 (H1 maml) |

## A.4 DATASET COMPOSITION DETAILS

Here we detail how we made our high diversity datasets. The method we used was taking the union as in (Tian et al., 2020) of different datasets. Fe used global labels (Wang et al., 2021) during training.

Here we outline what the acronyms mean in tables 12 and 11:

- HDBi stands for High-Diversity Benchmark number $i$.

- MIO stands for combining MiniImagenet (Vinyals et al., 2017) and Omniglot (Lake et al., 2015).

- MICOD stands for combining MiniImagenet (Vinyals et al., 2017), Cifar-fs (Bertinetto et al., 2019), Omniglot (Lake et al., 2015), and Delaunay (Gontier et al., 2022).

- AFDO stands for combining fgvcAircraft (Maji et al., 2013), vggFlower (Nilsback & Zisserman, 2006), Delaunay (Gontier et al., 2022), and Omniglot (Lake et al., 2015).

- AFTO stands for combining fgvcAircraft (Maji et al., 2013), vggFlower (Nilsback & Zisserman, 2006), describableTextures (Cimpoi et al., 2013), and Omniglot (Lake et al., 2015).

- CADO stands for combining Cifar-fs (Bertinetto et al., 2019), FGVCAircraft (Maji et al., 2013), Delaunay (Gontier et al., 2022), and Omniglot (Lake et al., 2015).

- CAVDO stands for combining Cifar-fs (Bertinetto et al., 2019), FGVCAircraft (Maji et al., 2013), VGGFlower (Nilsback & Zisserman, 2006), DescribableTextures (Cimpoi et al., 2013), Omniglot (Lake et al., 2015).

- MICOVA stands for combining MiniImagenet (Vinyals et al., 2017), Cifar-fs (Bertinetto et al., 2019), Omniglot (Lake et al., 2015), VGGFlower (Nilsback & Zisserman, 2006), FGVCAircraft (Maji et al., 2013).

- MDS stands for Meta-Dataset (Triantafillou et al., 2019).

- dtd stands for Describable Textures Dataset (Cimpoi et al., 2013).

- VGGFlower is the alternative name for fgvcFlower (Nilsback & Zisserman, 2006).

- vggair stands for combining VGGflower (Nilsback & Zisserman, 2006) and fgvcAircraft (Maji et al., 2013).

- vggdtd stands for combining VGGflower (Nilsback & Zisserman, 2006) and DTD (Cimpoi et al., 2013).

### A.4.1 FORMAL DIVERSITIES OF DATASETS

In this section, we present the formal diversities computed on the datasets we studied. We used the Task2Vec diversity coefficient described in section 2 and Miranda et al. (2022). We divide the results into datasets with formally low and high diversity. We do the division of low vs high at $0.146$ Task2Vec diversity (using a pre-trained (pt) Resnet18 (Renset18 pt)), because that was approximately the average diversity. We describe the dataset composition in the supplementary section A.4.

Table 11: **Shows the low diversities of datasets used for analysis using the Task2Vec diversity coefficient – using the train split with 95% confidence intervals.** We use a Resnet18 and Resnet34 pre-trained (pt) on ImageNet as the backbone to calculate the Fisher Information Matrix (FIM) needed for the Task2Vec task embeddings for the Diversity Coefficient.

| Dataset | Diversity (Resnet18 pt) | Diversity (Resnet34 pt) |
|---|---|---|
| cifar-fs | $0.106 \pm 0.00166$ | $0.0890 \pm 0.00199$ |
| fc100 | $0.107 \pm 0.00149$ | $0.0903 \pm 0.00389$ |
| mini-imagenet | $0.119 \pm 0.00213$ | $0.102 \pm 0.00163$ |
| tiered-imagenet | $0.124 \pm 0.00219$ | $0.105 \pm 0.00161$ |
| aircraft | $0.110 \pm 0.00127$ | $0.0932 \pm 0.00109$ |
| flower | $0.138 \pm 0.00288$ | $0.117 \pm 0.00234$ |
| dtd | $0.129 \pm 0.00227$ | $0.111 \pm 0.00228$ |
| delaunay | $0.128 \pm 0.00268$ | $0.1078 \pm 0.00196$ |
| cubirds | $0.120 \pm 0.00161$ | $0.104 \pm 0.00149$ |
| vggair | $0.141 \pm 0.00131$ | $0.120 \pm 0.00129$ |
| vggdtd | $0.135 \pm 0.00105$ | $0.119 \pm 0.00107$ |

## A.5 FURTHER TESTING OF THE DIVERSITY COEFFICIENT

### A.5.1 VALIDATING THE TASK2VEC TASK EMBEDDINGS USED IN THE DIVERSITY COEFFICIENT

In this section, we further test if the Task2Vec task embeddings distances cluster in a semantically meaningful way in our dataset MIO. This test is important because if the Task2Vec embeddings used to compute the diversity coefficient have the structure we'd expect, then it makes the diversity coefficient itself more trustworthy. The MIO dataset was created by combining the MiniImagenet Omniglot. Therefore, if Task2Vec is a valid embedding for tasks, we would expect three modes for our histogram: 1. One mode for the distances between tasks generated from MiniImagenet and MiniImagnet 2. Another mode for distances between tasks generated from Omniglot and Omniglot 3. And the last mode for distances between tasks generated from MiniImagenet and Omniglot That is indeed what is seen as shown in figure 2

One interesting observation is that the average distance between Task2Vec embeddings (i.e. the diversity coefficient) is larger for smaller networks.

Table 12: **Shows the high diversities of datasets used for analysis using the Task2Vec diversity coefficient – using the train split with 95% confidence intervals.** We use a Resnet18 and Resnet34 pre-trained (pt) on ImageNet as the backbone to calculate the Fisher Information Matrix (FIM) needed for the Task2Vec task embeddings for the Diversity Coefficient. MDS stands for Meta-DataSet.

| Dataset | Diversity (resnet18pt) | Diversity (resnet34pt) |
|---|---|---|
| omniglot | $0.167 \pm 0.00579$ | $0.139 \pm 0.00387$ |
| mio | $0.188 \pm 0.00416$ | $0.161 \pm 0.00351$ |
| hdb4-micod | $0.174 \pm 0.00420$ | $0.154 \pm 0.00381$ |
| hdb6-afdo | $0.179 \pm 0.00255$ | $0.155 \pm 0.00218$ |
| hdb7-afto | $0.186 \pm 0.00276$ | $0.146 \pm 0.00233$ |
| hdb8-cado | $0.173 \pm 0.00278$ | $0.153 \pm 0.00236$ |
| hdb9-cavdo | $0.177 \pm 0.00256$ | $0.139 \pm 0.00199$ |
| hdb10-micova | $0.170 \pm 0.00262$ | $0.137 \pm 0.00214$ |
| hdb12-fvfo | $0.181 \pm 0.00261$ | $0.150 \pm 0.00223$ |
| hdb13-mcfo | $0.166 \pm 0.00276$ | $0.159 \pm 0.00249$ |
| hdb14-dcmo | $0.179 \pm 0.00295$ | $0.150 \pm 0.00244$ |
| mds | $0.173 \pm 0.00282$ | $0.149 \pm 0.00252$ |

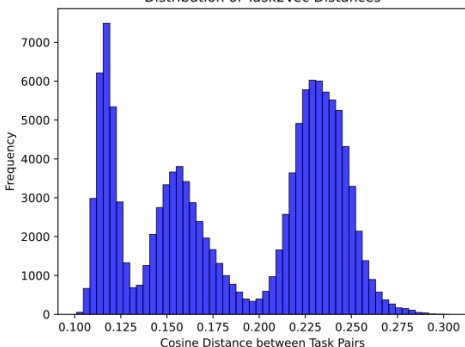 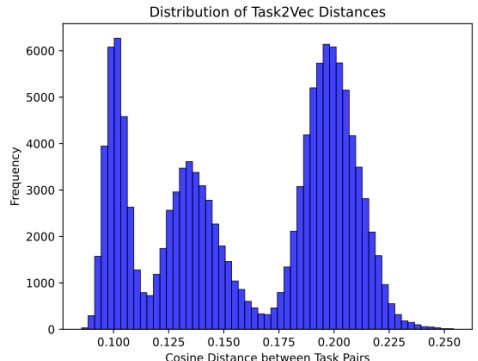

Figure 2: **The Task2Vec distances the histogram cluster in a way that reflect the semantic information of the union of the MiniImagenet and Omniglot (MIO) training datasets.** Left plot show the histogram of the cosine distance between Task2Vec embeddings made using a Resnet18 backbone pre-trained on Imagenet. Right show the same, but when using Resnet34. The meta-batch size was 500 meaning we used 500 tasks to compute these histograms. The diversity coefficients are $0.188 \pm 0.00416$, $0.161 \pm 0.00351$ for the LHS and RHS plots.

### A.6 DISCUSSION (CONT.)

We'd also like to remark the trade-offs between pre-training and meta-learning (MetaL) that Bronskill et al. (2021) articulates clearly – especially given the evidence we present countering the prevailing narrative that advocates for pre-training (PT) and transfer learning methods. The selection between PT vs MetaL strategies should be guided by: the available data, computational resources, and the application's specific requirements. For singular task types with ample data and no computational or temporal constraints, fine-tuning within transfer learning may suffice. Conversely, meta-learning would be more appropriate in scenarios requiring the acquisition of diverse tasks with sparse data on resource-limited devices, or in continual or online learning environments. In addition, we provide a novel perspective where we show formally diverse datasets are a scenario when meta-learning methods are marginally better than pre-training methods.

A potential drawback of our work could be our focus on mainly comparing pre-training (PT) against MAML, instead of considering a wider set of meta-learning algorithms. Our justification is as follows: The current narrative (Tian et al., 2020; Chen et al., 2019; 2020; Dhillon et al., 2019; Huang & Tao, 2019) implies that PT can beat *any* meta-learning algorithm. We'd like to emphasize the word *any*, because it implicates a "for all" quantifier. Therefore, to counter the current narrative, we only need

to provide evidence against it (and thus show it is likely false) by considering a *single* meta-learning algorithm. Therefore, if *PT cannot even beat MAML* – the simplest of meta-learning algorithms – it's good evidence against the current narrative. Therefore, we only need a *single* meta-learning algorithm to support our conclusions. In addition, memory efficient meta-learning (Bronskill et al., 2021) demonstrated that other meta-learning algorithms can match pre-trained models. However, as we explained in the related work section, our contributions are novel, complementary and different from Bronskill et al. (2021) does because: 1. We contextualize our claims in a data-centric framework using formal diversities over an extensive set of formally diverse datasets, 2. Our analysis goes beyond using confidence intervals and reports the effect size, a method we justify in section 3, and, 3. Our novel analysis demonstrates that meta-learning (via MAML) and pre-training can be (marginally) separated in performance when considering the formal diversity of the dataset. In addition, it is reasonable to expect, given the memory efficient results (Bronskill et al., 2021), that when using their memory efficient methods that a similar trend with other meta-learning algorithms would be observed – especially given we already showed an initial separation between PT and MAML. Furthermore, our experiments were are extensive over a large set of formally diverse datasets.

Another potential drawback of our work could be the use of the arbitrary 1% thresholds for our decision rule in Section 3. In machine learning, it is not uncommon to accept papers due to 1% differences. We cite this ICCV 2021 paper (Li et al., 2021) which gives the performance variance of common models on meta-dataset in table 1, which commonly ranges from $0.5$ to $1.0$. We also cite this ECCV 2020 paper (Rodríguez et al., 2020) that provides the variance for MiniImageNet, where the standard deviations range around $\approx 0.8$. However, we'd like to underscore that we **do not** rely solely on this 1% cutoff to interpret our experiments. We also report the raw effect size (and test accuracy) and use the classically accepted ranges for what is considered small effect size (Andrade, 2020). However, there is no silver bullet for statistical analysis. All of them have assumptions (e.g. CIs, effect size are best for normally distributed data), and some notion of arbitrary values (e.g. p-values, 95%-confidence intervals, effect size ranges, our 1% threshold, etc.) are always chosen to give meaning to the results. However, one can avoid confirmation bias by choosing the statistical method before the analysis of the experiments is done – which we do. In addition, our main rationale to chose effect size is that one can't manipulate (deliberately or accidentally) the sample size to have the decision rule match our preconceived assumptions – unlike the p-value in t-tests or confidence intervals where it has been an issue noted here (Lin et al., 2013) in the large sample size regime. Therefore, we attempted to protected our interpretations from confirmation bias.

Another criticism of our work could be the lack of theoretical analysis. One reason we choose not to do theoretical analysis is that it is often difficult to give non-vacuous bounds in theory. Though some progress has been done here Chen et al. (2021) but does not aim to separate pre-training methods vs meta-learning methods. However, our experiments have good theoretical motivation inspired from Wang et al. (2021) and align with conjectures we explain in detail in section A.7.2.

The challenges in vectorization of MAML and meta-learning algorithms in general stems from the arises because of the task are different across a meta-batch, so the support set has different arbitrary labels across tasks. Therefore, vectorization is not straightforward without custom CUDA implementations. However, instead of vectorizing, one could use the memory-efficient meta-learning strategy (Bronskill et al., 2021) to speed up MAML. This is an argument in favor of meta-learning given this new possible memory optimization. We leave this promising direction for future work but conjecture this will make MAML competitive against pre-training given our results and their results (Bronskill et al., 2021).

Most of our experiments are on small models but hypothesize they are all generalize. There is a debate about emergence in large language models, however, we hypothesize our results generalize to all size models. We hypothesize this because the observation of emergence is highly dependent on the metric and the sharp unpredictable jumps go away when using smooth metrics (Schaeffer et al., 2023).

An additional benefit of using the effect size is that it also protects the researchers from confirmation bias. For example, the researcher cannot deliberately choose a sample/batch size to fit pre-conceived assumptions.

### A.6.1 WHY AND WHEN DOES DIVERSITY MATTER?

We conjecture two main reasons why diversity matters and explain our rationale:

1. **Conjecture 1: Diversity matters because it enables learning-to-learn (proxy for General Intelligence).** This is the main conjecture we provide evidence in this paper. The main argument is that if there is high diversity, it means there are many tasks in the dataset. Therefore, for the model to do well, it has to do well on all tasks. One way to do it is by learning-to-learn and therefore transfer when challenged with solving a new task. An alternative would be memorizing all the tasks.

2. **Conjecture 2: Diversity matters because it increases changes that training set covers test set.** Diversity is a formalization of coverage – it aims to be the effective (average) number of tasks in a dataset. Therefore, the higher the diversity, the more tasks a dataset has. This (might) increase the probability that the training set covers the test set and improves performance. This exploration of this conjecture is left for future work.

## A.7 MOTIVATION

This work is inspired by three ideas/questions: 1. Does explicitly train to "learn to learn" (i.e., meta-learn) improve the performance of a machine learning algorithm? 2. What is a data-centric inductive bias that might explain when explicit meta-learning methods are needed? 3. Previous theoretical results hint that pre-training with all the data upper-bounds the meta-learning episodic loss, therefore, might this be the reason pre-training be slightly worse with imperfect settings? (e.g. imperfect optimization and limited data). We proceed to explain the latter two in more detail in this section.

### A.7.1 WHAT IS THE RIGHT INDUCTIVE BIAS FOR THE APPLICATION OF META-LEARNING?

Our work is motivated by the conjecture that the appropriate inductive bias for meta-learning is when the intrinsic diversity of a dataset is high. In other words, the distance between tasks sampled from a dataset is often high, i.e. a large variation of tasks is present. This is what the diversity coefficient is designed to measure (Miranda et al., 2022). The reasoning, behind this conjecture, is the following: 1. By definition of the problem – solving tasks from a high diversity dataset – we have tasks sampled from the dataset have large changes/distances 2. Therefore, a model that learns to adapt/learn/change might experience an advantage, because it has autonomously learns to change to these changing tasks. Consequently, if the tasks exhibit high variability/diversity, a meta-learning model may be the preferred choice.

The conjecture that meta-learning surpasses pre-training methods finds support in our empirical results 4, because the effect size is in favor of MAML on average across a wide variety of formally diverse datasets. However, intriguingly, pre-training methods outperform on low diversity datasets. This observation may clarify the prevailing narrative favoring pre-training methods. Pre-training methods might be better for lower diversity datasets. We hypothesize MAML may be "meta-overfitting" (Miranda et al., 2021) on such datasets, unlike pre-training methods with a fixed embedding that might have a lower (meta) variance.

Our problem/data-centric approach to meta-learning is inspired by applying Marr's level of analysis (Hamrick & Mohamed, 2020; Marr, 1982) to few-shot learning. Marr emphasized the importance of understanding the computational problem being solved and not only analyzing the algorithms or hardware that attempts to solve them. An example Marr gives is marveling at the rich structure of bird feathers without also understanding the problem they solve: flight. Similarly, there has been an analysis of MAML models and transfer learning without putting the problem such models should solve into perspective (Raghu et al., 2020; Tian et al., 2020). Therefore, in this work, we hope to clarify some of these results by partially placing the current state of affairs in meta-learning from a problem-centric view. We do this by computing the formal diversity of a dataset using the diversity coefficient.

A.7.2 THEORETICAL MOTIVATION

Our work is also inspired by the theory from Wang et al. (2021), which theoretically shows the loss of pre-training on all the data upper bounds the episodic meta-learning loss.

More formally, for a fixed feature embedding model $\psi_\theta(x)$ with weights $\theta$:

$$\mathbb{E}_{(X,Y)\in Q}\left[L_{ce}(W[Y],(\psi_\theta(X),Y))\right] \leq \mathbb{E}_{(x,y)\in D(Q)}\left[l_{ce}(W\psi_\theta(x),y)\right] \tag{4}$$

where $Q = \{(x_i,y_i)\}_{i=1}^{n_Q}$ is a standard few-shot learning query set, $L_{ce}$ is the empirical risk of the learner over few-shot tasks using the cross-entropy loss (i.e. on the support or a query set), $X$ is a dataset of raw input values $x$ e.g. raw images, $Y$ is a dataset of target labels e.g. labels for the images, $\psi_\theta(X) = \{\psi_\theta(x) \mid x \in X\}$ the embedded few-shot task/dataset, and $D(Q)$ denotes the union of all query tasks from the source dataset e.g. union of all few-shot learning of MiniImagenet.

Equation 4 therefore implies that using a fixed embedding method, that a pre-trained model using all the tasks upper bounds the true meta-learning loss we wish to minimize for a model to "learn to learn". The main caveat is that this only applies for a fixed embedding model, so if the left-hand side uses a MAML model and the right-hand side uses a pre-trained model, then the above inequality doesn't apply. However, it provides good heuristics for our approach: 1. Get an extremely large dataset, 2. Train both models to convergence, ideally zero train-loss, 3. then compare them. The above suggests the difference should be small, which is, which is in line with our main contribution. In addition, the effect size is negative which suggests MAML is better, as one might conjecture using equation 4. In addition, as the meta-train set encompasses all possible tasks, we conjecture there is no difference between meta-leanring algorithms and pre-training trained on a union of all the data.

A.8 MAML EXPERIMENTS ON GPT-2

Extending MAML for language modelling is a challenging task. Large language models (LLMs) including GPT-2 are typically trained in a supervised learning setting where the model is trained to predict the token coming after each token in a sentence(Radford et al., 2019). This does not naturally translate to a $k$-shot learning task which MAML was intended for. In particular, we note that the vocabulary size for each token is in the order of tens of thousands (50257 for GPT-2) which is a lot bigger than the few thousand (1623 for Omniglot(Lake et al., 2015)) classes MAML is typically used for. Additionally, each token does not have an equal number of instances in the language. Finally and most importantly, training a model to chose between a few classes given example occurrences of those classes, and comparing it against a model trained to predict one class out of 50257 is an apples to oranges comparison.

We however note the primary motivation of the paper to establish the performance of "learning a task" against "learning to learn the same task". Hence instead of using examples of specific target classes as the support set as is usually done for MAML, we use example sequences as the support set and example sequences as the query set. An initial idea is for each batch size of size $b$, we can train the model to learn from the first $b/2$ examples and predict the next $b/2$ correctly. However, since the batches are independent, we don't give the model a chance to learn from context and this implementation of MAML reduces to a harder to optimize version of the usual supervised learning setting.

We solve this issue by separating the support and query sets within examples in a batch instead of within batches. That is, if the token size of the model is $t$, we split each example of $t$ tokens into support and query sets. We make the first $t/2$ tokens of each example as the support set and the next $t/2$ tokens as the query set. Hence given a token size $t$, we train the model to learn to predict the last $t/2$ tokens based on the first $t/2$ tokens. Formally, for each example in the batch, we perform an inner loop optimization on the cross-entropy loss of next-token predictions for the first $t/2$ tokens. We then perform the outer loop optimization on the cross-entropy loss of next-token predictions using the obtained model on the last $t/2$ tokens.

Evaluation comparison between the two training techniques is done by following a similar approach of inner loop optimization using the first half tokens and reporting accuracy values on the second half post inner loop optimization.

### A.9    FAIR COMPARISON

Unlike previous, we ensure fair comparison between pre-training vs MAML by using a consistent neural network architecture, optimizer, and all models trained to convergence.

**Architecture:** We only compared pre-training vs MAML when they **both** had the same architecture. When we used a ResNet we used the one described in Tian et al. (2020).

**Optimization:** We only compared pre-training vs MAML when they **both** had the same optimization and scheduling rate. We used the Adam optimizer for all experiments except for GPT2 and Resnet50 on Meta-DataSet (MDS) where we used Adafactor with default hyperparameters. We did this because Adafactor has a setting in the Fairseq that requires no hyperparameter search and since Meta-DataSet is a large we scale dataset. It took us about 1 month to train on MDS with Resnet50. In addition, previous work demonstrated Adafactor can be fast 1 order of magnitude faster (speedup of 2 hours to 39 hours) than Adam with hyperparameter search when training transformer models (Miranda et al., 2023).

**Training to Convergence:** We show how our models were trained by providing some sample learning curves for pre-training and MAML in the following figures 7, 8, 5, 6, 3, 4 .

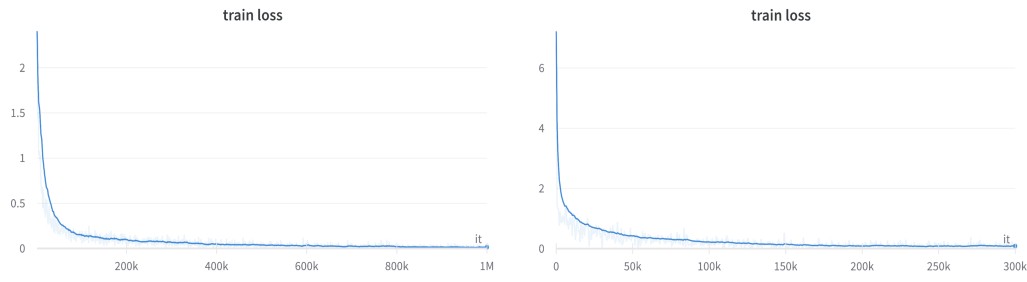

Figure 3: Plot showing convergence of Resnet12 on a high-diversity benchmark (MICOD). The left-most plot depicts the training loss curve for the pre-training algorithm, and the rightmost plot depicts the training loss curve for MAML.

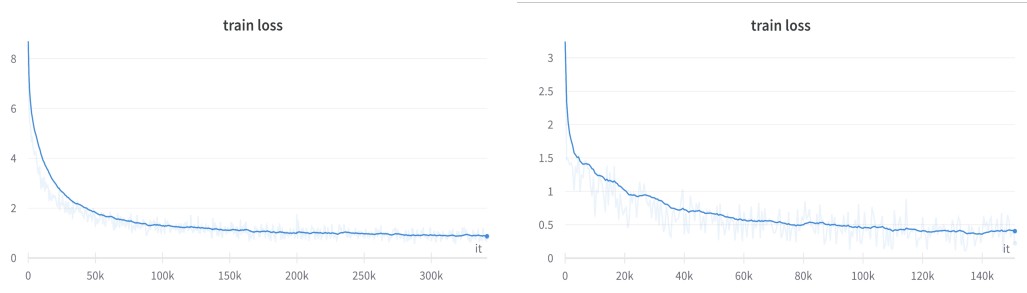

Figure 4: Plot showing convergence of Resnet50 on a high-diversity benchmark Meta-DataSet (MDS) (Triantafillou et al., 2019). The left-most plot depicts the training loss curve for the pre-training algorithm, and the rightmost plot depicts the training loss curve for MAML.

### A.10    SUMMARY OF CI DECISION RESULTS

When comparing PT (pre-training) and MAML using confidence intervals, our experiments indicate that MAML and PT tend to perform equivalently under high-diversity benchmarks, while MAML and PT perform differently (either MAML outperforming or underperforming PT) under lower-diversity benchmarks.

Figure 11 shows how average MAML(5,10) performs better than PT. This supports our main hypothesis because 1. MAML is better than PT in the high diversity regime but 2. The difference is marginal, as shown by the confidence intervals being close.

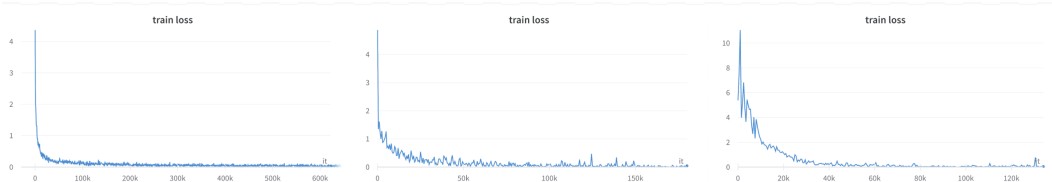

Figure 5: Plot showing convergence of Resnet12 on a low-diversity benchmark (fc100). The left-most plot depicts the training loss curve for the pre-training algorithm, the center plot depicts the training loss curve for first-order MAML, and the rightmost plot depicts the training loss curve for higher-order MAML.

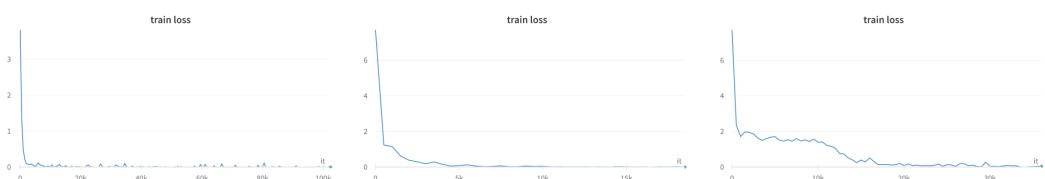

Figure 6: Plot showing convergence of Resnet12 on a low-diversity benchmark (aircraft). The left-most plot depicts the training loss curve for the pre-training algorithm, the center plot depicts the training loss curve for first-order MAML, and the rightmost plot depicts the training loss curve for higher-order MAML.

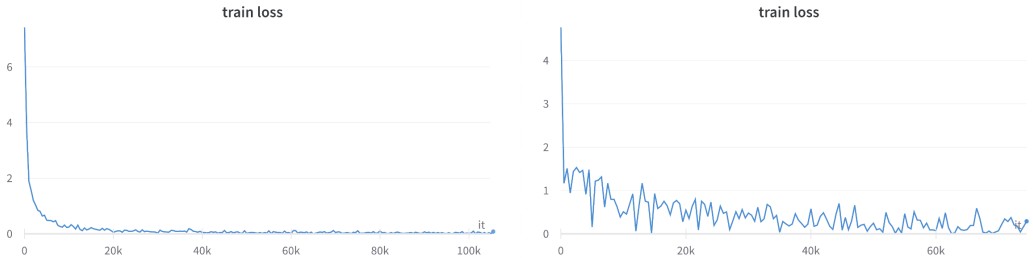

Figure 7: Plot showing convergence of Resnet12 on a high-diversity benchmark (hdb8-cado).The left plot depicts the training loss curve for the pre-training algorithm and the right plot depicts the training loss curve for MAML.

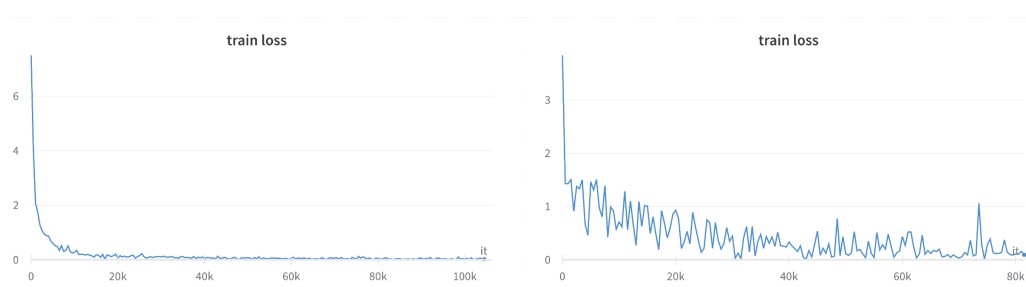

Figure 8: Plot showing convergence of Resnet12 on a high-diversity benchmark (hdb9-cavdo). The left plot depicts the training loss curve for the pre-training algorithm and the right plot depicts the training loss curve for MAML.

Table 13: **Results of performance comparison between pre-training and MAML using confidence intervals for high-diversity benchmarks.** These performance comparison experiments were conducted using a batch size of 300. The summary for the decision counts is as follows: we failed to reject the null hypothesis H0 (no difference) 8 times and we rejected the null hypothesis in favor of the MAML alternative 2 times (once for MAML5 and once for MAML10). The diversity for them is high, as shown in table 12.

| Dataset | pt vs maml5 CI decision | pt vs maml10 CI decision |
|---|---|---|
| hdb6-afdo | H0 no diff | H0 no diff |
| hdb7-afto | H0 no diff | H0 no diff |
| hdb8-cado | H0 no diff | H0 no diff |
| hdb9-cavdo | H0 no diff | H0 no diff |
| hdb10-micova | H1 maml5 | H1 maml10 |

Table 14: **Results of performance comparison between pre-training and MAML using confidence intervals for high-diversity benchmarks with a 1% overlap threshold.** These performance comparison experiments were conducted using a batch size of 300. The summary for the decision counts is as follows: we failed to reject the null hypothesis H0 (no difference) 8 times and we rejected the null hypothesis in favor of the MAML alternative 2 times (once for MAML5 and once for MAML10). The diversity for them is high, as shown in table 12.

| Dataset | pt vs maml5 CI decision (1% overlap) | pt vs maml10 CI decision (1% overlap) |
|---|---|---|
| hdb6-afdo | H0 no diff | H0 no diff |
| hdb7-afto | H0 no diff | H0 no diff |
| hdb8-cado | H0 no diff | H0 no diff |
| hdb9-cavdo | H0 no diff | H0 no diff |
| hdb10-micova | H1 maml5 | H1 maml10 |

### A.11 L2 MODEL NORMS AND VALIDATION LOSS CURVES SUGGEST THAT MAML HAS LESS META-OVERFITTING THAN PT

We demonstrate evidence that may suggest that MAML has less overfitting than PT, both via MAML and PT validation loss curves (see Figures 9 and 10), as well as the L2 model norms of trained MAML and PT models (see Tables 22, 23, 24).

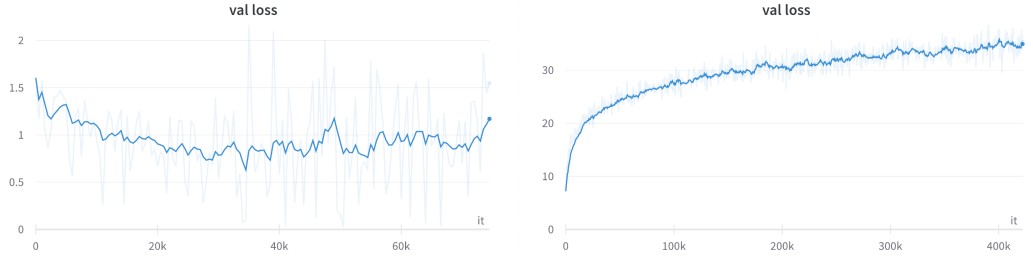

Figure 9: **On a high-diversity dataset, the validation loss of MAML stays relatively unchanged over time, while the validation loss of PT increases over time, suggesting that MAML has less meta-overfitting than PT.** The left plot depicts the validation loss curve for the MAML algorithm and the right plot depicts the validation loss curve for the PT algorithm, both on the high-diversity hdb8-cado dataset.

Table 15: **Results of performance comparison between pre-training and (fo) MAML using confidence intervals for low-diversity benchmarks.** These performance comparison experiments were conducted using a batch size of 300. The summary for the decision counts is as follows: we failed to reject the null hypothesis H0 (no difference) 11 times, we rejected the null hypothesis in favor of the PT alternative 7 times, and rejected the null hypothesis in favor of the MAML alternative 6 times (MAML5 accounted for 3 of these rejections while MAML10 accounted for 3). The diversity for them is low, as shown in table 11.

| Dataset | pt vs maml5 CI decision | pt vs maml10 CI decision |
|---|---|---|
| aircraft | H1 no diff | H1 no diff |
| flower | H1 pt | H1 pt |
| dtd | H1 pt | H1 pt |
| delaunay | H1 pt | H1 pt |
| cubirds | H1 maml5 | H1 maml10 |
| cifar-fs | H1 maml5 | H1 maml10 |
| fc100 | H1 maml5 | H1 maml10 |
| mini-imagenet | H1 pt | H0 no diff |
| omniglot | H0 no diff | H0 no diff |
| tiered-imagenet | H0 no diff | H0 no diff |
| vggair | H0 no diff | H0 no diff |
| vggdtd | H0 no diff | H0 no diff |

Table 16: **Results of performance comparison between pre-training and (fo) MAML using confidence intervals for low-diversity benchmarks with a 1% overlap threshold.** These performance comparison experiments were conducted using a batch size of 300. The summary for the decision counts is as follows: we failed to reject the null hypothesis H0 (no difference) 14 times, we rejected the null hypothesis in favor of the PT alternative 7 times, and rejected the null hypothesis in favor of the MAML alternative 3 times (MAML5 accounted for 1 of these rejections while MAML10 accounted for 2). The diversity for them is low, as shown in table 11.

| Dataset | pt vs maml5 decision (1% overlap) | pt vs maml10 decision (1% overlap) |
|---|---|---|
| aircraft | H0 no diff | H0 no diff |
| flower | H1 pt | H1 pt |
| dtd | H1 pt | H1 pt |
| delaunay | H1 pt | H1 pt |
| cubirds | H1 maml5 | H1 maml10 |
| cifar-fs | H0 no diff | H1 maml10 |
| fc100 | H0 no diff | H0 no diff |
| mini-imagenet | H1 pt | H0 no diff |
| omniglot | H0 no diff | H0 no diff |
| tiered-imagenet | H0 no diff | H0 no diff |
| vggair | H0 no diff | H0 no diff |
| vggdtd | H0 no diff | H0 no diff |

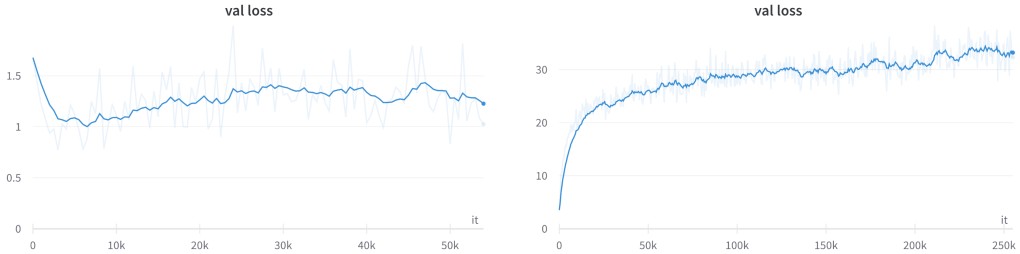

Figure 10: **On a low-diversity dataset, the validation loss of MAML stays relatively unchanged over time, while the validation loss of PT increases over time, suggesting that MAML has less meta-overfitting than PT.** The left plot depicts the validation loss curve for the MAML algorithm and the right plot depicts the validation loss curve for the PT algorithm, both on the low-diversity DTD dataset.

Table 17: **Results of performance comparison between pre-training and (ho) MAML using confidence intervals for low-diversity benchmarks.** These performance comparison experiments were conducted using a batch size of 300. The summary for the decision counts is as follows: we failed to reject the null hypothesis H0 (no difference) 2 times, we rejected the null hypothesis in favor of the PT alternative 10 times, and rejected the null hypothesis in favor of the MAML alternative 8 times (MAML5 accounted for 4 of these rejections while MAML10 accounted for 4). The diversity for them is low, as shown in table 11.

| Dataset | pt vs maml5 CI decision | pt vs maml10 CI decision |
|---|---|---|
| aircraft | H1 maml5 | H1 maml10 |
| flower | H1 pt | H1 pt |
| dtd | H1 pt | H1 pt |
| delaunay | H1 pt | H1 pt |
| cubirds | H1 maml5 | H1 maml10 |
| cifar-fs | H1 maml5 | H1 maml10 |
| fc100 | H1 maml5 | H1 maml10 |
| mini-imagenet | H0 no diff | H0 no diff |
| omniglot | H1 pt | H1 pt |
| tiered-imagenet | H1 pt | H1 pt |

Table 18: **Results of performance comparison between pre-training and (ho) MAML using confidence intervals for low-diversity benchmarks with a 1% overlap threshold.** These performance comparison experiments were conducted using a batch size of 300. The summary for the decision counts is as follows: we failed to reject the null hypothesis H0 (no difference) 6 times, we rejected the null hypothesis in favor of the PT alternative 6 times, and rejected the null hypothesis in favor of the MAML alternative 8 times (MAML5 accounted for 4 of these rejections while MAML10 accounted for 4). The diversity for them is low, as shown in table 11.

| Dataset | pt vs maml5 CI decision (1% overlap) | pt vs maml10 CI decision (1% overlap) |
|---|---|---|
| aircraft | H1 maml5 | H1 maml10 |
| flower | H0 no diff | H1 pt |
| dtd | H1 pt | H1 pt |
| delaunay | H1 pt | H1 pt |
| cubirds | H1 maml5 | H1 maml10 |
| cifar-fs | H1 maml5 | H1 maml10 |
| fc100 | H1 maml5 | H1 maml10 |
| mini-imagenet | H0 no diff | H0 no diff |
| omniglot | H0 no diff | H0 no diff |
| tiered-imagenet | H0 no diff | H1 pt |

Table 19: 1% effect sizes for performance comparison between pre-training and (fo) MAML for low-diversity benchmarks.

| Dataset | pt vs maml5 1% ES | pt vs maml10 1% ES |
|---|---|---|
| aircraft | 0.100 | 0.109 |
| flower | 0.171 | 0.195 |
| dtd | 0.122 | 0.125 |
| delaunay | 0.106 | 0.122 |
| cubirds | 0.135 | 0.133 |
| cifar-fs | 0.114 | 0.113 |
| fc100 | 0.121 | 0.117 |
| mini-imagenet | 0.123 | 0.117 |
| omniglot | 0.789 | 0.727 |
| tiered-imagenet | 0.112 | 0.113 |
| vggair | 0.059 | 0.059 |
| vggdtd | 0.056 | 0.056 |

Table 20: 1% effect sizes for performance comparison between pre-training and (ho) MAML for low-diversity benchmarks.

| Dataset | pt vs maml5 1% ES | pt vs maml10 1% ES |
|---|---|---|
| aircraft | 0.094 | 0.100 |
| flower | 0.201 | 0.204 |
| dtd | 0.125 | 0.126 |
| delaunay | 0.116 | 0.115 |
| cubirds | 0.145 | 0.145 |
| fc100 | 0.113 | 0.113 |
| cifar-fs | 0.118 | 0.113 |
| mini-imagenet | 0.120 | 0.123 |
| omniglot | 0.655 | 0.762 |
| tiered-imagenet | 0.116 | 0.080 |

Table 21: 1% effect sizes for performance comparison between pre-training and (fo) MAML for high-diversity benchmarks.

| Dataset | pt vs maml5 1% ES | pt vs maml10 1% ES |
|---|---|---|
| hdb6-afdo | 0.057 | 0.059 |
| hdb7-afto | 0.050 | 0.051 |
| hdb8-cado | 0.051 | 0.053 |
| hdb9-cavdo | 0.054 | 0.054 |
| hdb10-micova | 0.056 | 0.057 |

Table 22: **The L2 norm of a trained first-order MAML model is less than the L2 norm of a trained PT model for each low-diversity benchmark, suggesting that MAML has less meta-overfitting than PT.**

| Dataset | L2 model norm (MAML) | L2 model norm (PT) |
|---|---|---|
| cifar-fs | 851.012 | 9813.269 |
| fc100 | 919.964 | 8302.170 |
| omniglot | 663.715 | 4676.182 |
| mini-imagenet | 930.304 | 5776.625 |
| tiered-imagenet | 926.896 | 11097.097 |

Table 23: **The L2 norm of a trained higher-order MAML model is less than the L2 norm of a trained PT model for each low-diversity benchmark, suggesting that MAML has less meta-overfitting than PT.**

| Dataset | L2 model norm (MAML) | L2 model norm (PT) |
|---|---|---|
| dtd | 2949.312 | 5548.194 |
| tiered-imagenet | 731.893 | 11097.097 |
| omniglot | 594.381 | 4676.182 |
| fc100 | 758.372 | 8302.170 |
| delaunay | 2810.343 | 4856.295 |
| aircraft | 1517.484 | 6144.078 |
| cifar-fs | 725.017 | 9813.269 |
| mini-imagenet | 752.201 | 5776.625 |
| cubirds | 3127.715 | 5252.014 |
| flower | 2333.556 | 7307.350 |

Table 24: **The L2 norm of a trained higher-order MAML model is less than the L2 norm of a trained PT model for each high-diversity benchmark, suggesting that MAML has less meta-overfitting than PT.**

| Dataset | L2 model norm (MAML) | L2 model norm (PT) |
|---|---|---|
| hdb6-afdo | 2426.827 | 7699.514 |
| hdb7-afto | 2598.023 | 3573.5355 |
| hdb8-afdo | 3441.040 | 8072.174 |
| hdb9-cavdo | 3997.130 | 7919.635 |
| hdb10-micova | 3810.151 | 7757.541 |

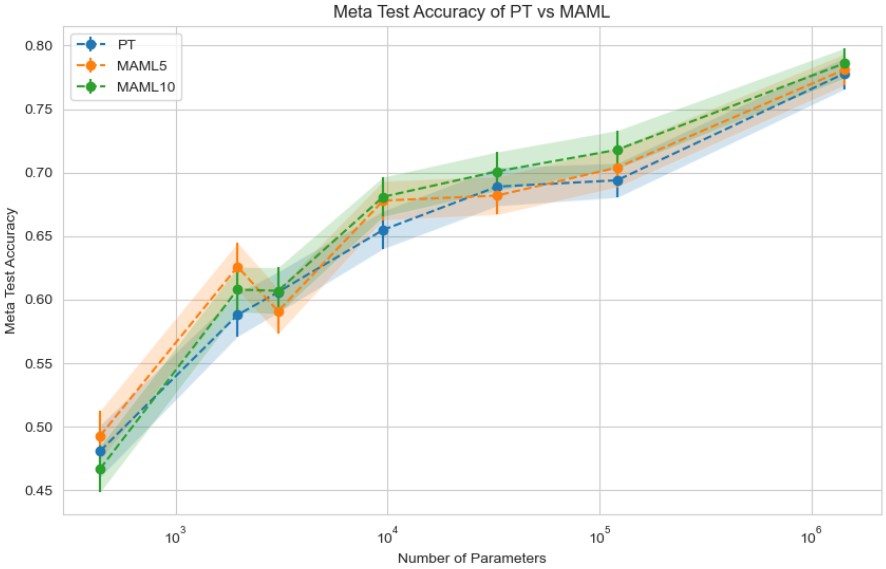

Figure 11: **Shows how meta-test accuracy between PT and MAML(5,10) intersects in the high diversity dataset MICOD.** However, on average MAML(5,10) performs better than PT. This supports our main hypothesis because: 1. MAML is better than PT in the high diversity regime but 2. The difference is marginal, as shown by the confidence intervals being close.

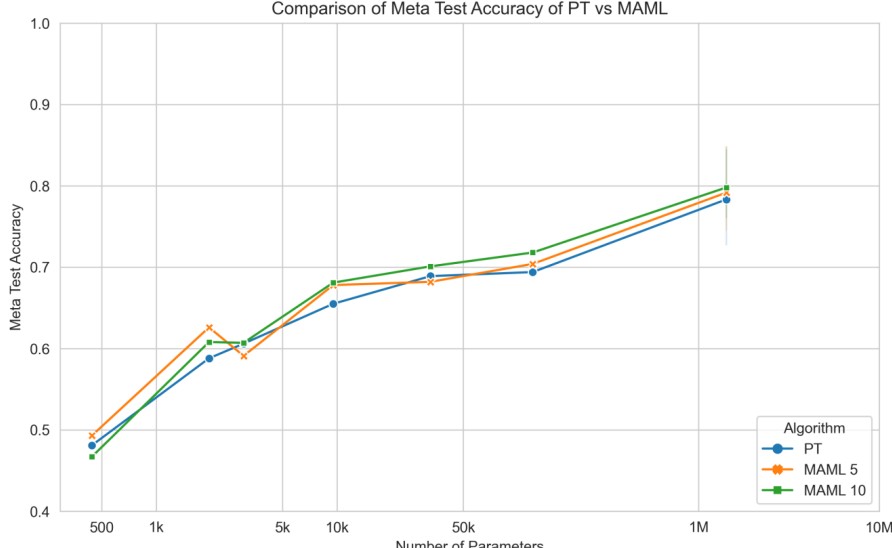

Figure 12: **MAML(5,10) outperforms PT in the high diversity dataset MICOD.** Same as Figure 11 but without confidence intervals. This supports our main hypothesis as it demonstrates that MAML is marginally better than PT in the high diversity regime.

