# OpenReview forum: "Is Pre-training Truly Better Than Meta-Learning?"
_ICLR.cc/2024/Conference — Submitted to ICLR 2024_

### Official Review · Reviewer_TVer · 2023-10-30

**Soundness:** 3 good
**Presentation:** 3 good
**Contribution:** 3 good
**Rating:** 5
**Confidence:** 4

**Summary:**

The paper conducts a comprehensive examination of the effectiveness of pre-trained (PT) models in comparison to Model Agnostic Meta-Learning (MAML) within the realm of few-shot learning. Despite prevailing beliefs suggesting the dominance of PT models, this study provides an unbiased comparison by employing identical architecture and ensuring complete model training. The research uses a robust statistical methodology, specifically the effect size (Cohen’s d), to discern the practical differences between PT and MAML models. A "diversity coefficient" metric is used to define the formal diversity of datasets. Key findings highlight that for datasets with low formal diversity, PT models slightly outperform MAML. Conversely, in scenarios with high formal diversity, MAML tends to be more effective. However, the magnitude of these differences is marginal, with an effect size less than 0.2. When evaluating the overall performance across various dataset diversities, neither method exhibits a clear advantage. The authors conclude that the inherent diversity of datasets is an important factor in determining the efficacy of PT versus MAML models.

**Strengths:**

- The paper is well written and the methodology is clearly described and framed.

- The empirical assessment is robust and comprehensive. By employing a diverse set of statistical techniques beyond standard benchmarks, the authors have elevated the quality of model evaluation.

- The methods are compared on a large variety of datasets which are clustered in terms of diversity.

**Weaknesses:**

- The paper is solid from the empirical point of view, presenting a large variety of results. However, it is somehow limited in terms of novelty as it does not introduce any new techniques or analyses.

- A significant limitation of the paper lies in its reliance on two methodologies, MAML and PT with fine-tuning restricted to the head, which may not represent the current best practices in the field. Firstly, while MAML is undoubtedly foundational in meta-learning, its relevance has waned over time. Contemporary advancements have introduced more efficient derivatives, such as MAML++ (Antoniou et al., 2018). Incorporating comparisons with these modern variants could have enriched the paper's insights. Secondly, the paper's approach to fine-tuning is notably narrow, focusing only on the head's parameter adjustments. Contrarily, cutting-edge methods today, like BiT (Kolesnikov et al., 20202), fine-tune the entirety of both body and head parameters, while others, like FiT (Shysheya et al., 2022), selectively adjust a subset of body parameters. A juxtaposition against these state-of-the-art techniques would have been insightful. These oversights are critical as the paper's primary conclusions might shift when evaluated against more contemporary, optimized methods.

- The presentation of data exclusively in tabular format, though beneficial for transparency, hinders a quick understanding of the trends. I recommend the authors to enhance data representation by incorporating visual aids, such as scatter plots. This would facilitate a more intuitive grasp of the data patterns. The tables could be conveniently relocated to the appendix to maintain thoroughness without overwhelming the main content.

- There are some formatting issues, e.g. (i) Table 6 should be within the body of the paper, (ii) in the text "Meta-Data set" should be replaced with "Meta-Dataset"



References
----------

Antoniou, A., Edwards, H., & Storkey, A. (2018). How to train your MAML. arXiv preprint arXiv:1810.09502.

Kolesnikov, A., Beyer, L., Zhai, X., Puigcerver, J., Yung, J., Gelly, S., & Houlsby, N. (2020). Big transfer (bit): General visual representation learning. In Computer Vision–ECCV 2020: 16th European Conference, Glasgow, UK, August 23–28, 2020, Proceedings, Part V 16 (pp. 491-507). Springer International Publishing.

Shysheya, A., Bronskill, J., Patacchiola, M., Nowozin, S., & Turner, R. E. (2022). Fit: Parameter efficient few-shot transfer learning for personalized and federated image classification. arXiv preprint arXiv:2206.08671.

**Questions:**

Do the authors believe that a comparison between various methods, such as MAML++, BiT, or FiT, is feasible? Additionally, would such comparisons yield consistent conclusions with the current findings?

Please refer to the "Weaknesses" session for other potential points of discussion.

---

> ### Author Response · Authors · 2023-11-22
> **Thank you for thoughtful review**
>
> Q1: The paper is solid from the empirical point of view, presenting a large variety of results. However, it is somehow limited in terms of novelty as it does not introduce any new techniques or analyses.
>
> A1: We respectfully differ. For instance, we have introduced a novel method of comparing two algorithms via the effect size, which we believe is more robust to sample size than p-values and confidence intervals (as described in Section 3.2). Furthermore, our paper also introduces a Task2Vec-based diversity coefficient that measures the task diversity of a given few-shot benchmark. We further present a detailed effect size-based statistical analysis of performance that gives substantial support to our claim that meta-learning outperforms pretraining if the data diversity is large enough.
>
> Q2: A significant limitation of the paper lies in its reliance on two methodologies, MAML and PT with fine-tuning restricted to the head, which may not represent the current best practices in the field. Firstly, while MAML is undoubtedly foundational in meta-learning, its relevance has waned over time. Contemporary advancements have introduced more efficient derivatives, such as MAML++ (Antoniou et al., 2018). Incorporating comparisons with these modern variants could have enriched the paper's insights. Secondly, the paper's approach to fine-tuning is notably narrow, focusing only on the head's parameter adjustments. Contrarily, cutting-edge methods today, like BiT (Kolesnikov et al., 20202), fine-tune the entirety of both body and head parameters, while others, like FiT (Shysheya et al., 2022), selectively adjust a subset of body parameters. A juxtaposition against these state-of-the-art techniques would have been insightful. These oversights are critical as the paper's primary conclusions might shift when evaluated against more contemporary, optimized methods.
>
> A2: We see this paper as a direct response to existing work that claims that meta-learning methods are worse than pre-training approaches, using equivalent methods to ours ( https://arxiv.org/abs/2003.11539 ). We propose the use of better experimental methods to allow a fair comparison between the techniques and show a reversal of what are now folklore conclusions. We note that the use of MAML as opposed to other meta-learning methods furthers our claim, since MAML is the simplest meta-learning approach. If MAML can outperform pre-training, better meta-learners should be able to do so too. Please find more details about this in the appendix, section A.6, paragraph 2.
>
> Q3: The presentation of data exclusively in tabular format, though beneficial for transparency, hinders a quick understanding of the trends. I recommend the authors to enhance data representation by incorporating visual aids, such as scatter plots. This would facilitate a more intuitive grasp of the data patterns. The tables could be conveniently relocated to the appendix to maintain thoroughness without overwhelming the main content.
>
> A3: Thanks for the suggestions. We have moved the detailed results to the appendix, leaving only tables providing a concise and easy to understand summary for our results in the main paper. We have additionally added plots indicating the spread of effect size for each of our 2 categories of diversity. Hoping this makes our results clearer and easier to understand.
>
> Q4: There are some formatting issues, e.g. (i) Table 6 should be within the body of the paper, (ii) in the text "Meta-Data set" should be replaced with "Meta-Dataset"
>
> A4: Thanks for pointing this out. We have fixed these issues in the revision.

---

> > ### Comment · Reviewer_TVer · 2023-11-22
> > **Answer to rebuttal**
> >
> > Thank you for the answer. After reading the rebuttal I still have some concerns that I have detailed below.
> >
> > 1) Regarding novelty. Most of the contributions listed in the rebuttal look to me like variations of techniques that already existed. For instance, in the answer you claim that Task2Vec is one of the contribution of the paper. However, as far as I know this was developed by Achille et al. (2019) and Miranda et al. (2022), so I am not sure if can count as a novel contribution. Moreover, if the primary contribution of your work is methodological, it might be beneficial to reorient the paper to emphasize how your proposed analytical techniques provide broad utility beyond existing methods. This reorientation could better highlight the novel aspects of your work.
> >
> > 2) Regarding comparison with other methods. My observation was mainly about comparing against more effective fine-tuning methods. As I mentioned in my review, fine-tuning only the head is a very simple baseline and there are much more effective methods out there (e.g. Kolesnikov et al. 2020, Shysheya et al. 2022). The main conclusion of the paper, that meta-learning outperforms fine-tuning, may not hold when state-of-the-art techinques come into play.
> >
> > Alessandro Achille, Michael Lam, Rahul Tewari, Avinash Ravichandran, Subhransu Maji, Stefano Soatto, and Pietro Perona. Task2Vec: Task Embedding for Meta-Learning. Technical report, 2019.
> >
> > Brando Miranda, Patrick Yu, Yu-Xiong Wang, and Sanmi Koyejo. The Curse of Low Task Diversity: On the Failure of Transfer Learning to Outperform MAML and Their Empirical Equivalence. arXiv, 2022. doi: 10.48550/arXiv.2208.01545
> >
> > Kolesnikov, A., Beyer, L., Zhai, X., Puigcerver, J., Yung, J., Gelly, S., & Houlsby, N. (2020). Big transfer (bit): General visual representation learning. In Computer Vision–ECCV 2020: 16th European Conference, Glasgow, UK, August 23–28, 2020, Proceedings, Part V 16 (pp. 491-507). Springer International Publishing.
> >
> > Shysheya, A., Bronskill, J., Patacchiola, M., Nowozin, S., & Turner, R. E. (2022). Fit: Parameter efficient few-shot transfer learning for personalized and federated image classification. arXiv preprint arXiv:2206.08671.

---

> > > ### Author Response · Authors · 2023-11-22
> > >
> > > Regarding the use of alternative methods for comparison: We emphasize the impossibility of achieving a fair comparison once we decide to go down that route. Can such a comparison actually ever be done, given that it's impossible to truly compare the best algorithms in practice? Since pre-training is more widely studied, it is not fair to compare the best pre-training and the best meta-learning algorithm. Owing to the bias in favour of pre-training in the literature, can we rule our confirmation bias if we use advanced algorithms for pre-training? Similarly, how do we argue our choice of MAML & pre-training wasn't cherry picked?
> > >
> > > We argue that MAML and head fine-tuning are the most similar to each other, without this type of bias, as well as the most fundamental approaches in both fields. This allows us to conduct the most "fair" comparison that we could. Our study aims to unravel the true difference between meta-leanring vs pre-training through a fair comparison with the validated hypothesis that data type plays a crucial factor as we demonstrated.

---

### Official Review · Reviewer_RzB7 · 2023-10-31

**Soundness:** 2 fair
**Presentation:** 1 poor
**Contribution:** 1 poor
**Rating:** 5
**Confidence:** 2

**Summary:**

The paper investigates the question of when meta-learning is suprior to pre-training, by using effect size (especially Cohen's d) as a statistical measure for comparison between the two approaches, on several few-shot classification benchmarks. The effect of task diversity in meta/pre-training is also considerred, which leads to findings that  pre-training (resp. meta-learning) tends to be suitable for low-diversity (resp. high-diversity) regime.

**Strengths:**

1. The paper uses an effect size to compare pre-training and meta-learning for the first time, which enables us to compare two methods quantitatively.
2. The paper empirically validates that the task diversity is a key property distinguishing pre-training and meta-learning, which is consistent with the motivation of meta-learning.

**Weaknesses:**

1. No novel methods are introduced in this paper, which itself is ok if the results are intriguing.
2. Presentation of the results is very poor. All results are just listed in tables, and there are no attempts to present the results in a comprehensible/impressive manner. Since I could not find any meaningful insight from the tables, I recommend adding more comprehensive figures which should be contributions of the paper.
3. The results of Cohen's d can be caused from (1) meta-training dataset, and/or (2) meta-test dataset, in addition to learning algorithms, but I could not find which factor causes the results of Cohen's d. In other words, which diversity of meta-training datasets or meta-testing datasets makes the difference between the two methods?

**Questions:**

See Weaknesses.

---

> ### Author Response · Authors · 2023-11-22
> **Thank you for thoughtful reviews**
>
> We thank Reviewer RzB7 for the insightful comments. Below is our response:
>
> Q1. No novel methods are introduced in this paper, which itself is ok if the results are intriguing.
>
> A1. We respectfully differ. For instance, we have introduced a novel method of comparing two algorithms via the effect size, which we believe is more robust to sample size than p-values and confidence intervals (as described in Section 3.2). Furthermore, our paper also introduces a Task2Vec-based diversity coefficient that measures the task diversity of a given few-shot benchmark. We further present a detailed effect size-based statistical analysis of performance that gives substantial support to our claim that meta-learning outperforms pretraining if the data diversity is large enough.
>
> Q2.  Presentation of the results is very poor. All results are just listed in tables, and there are no attempts to present the results in a comprehensible/impressive manner. Since I could not find any meaningful insight from the tables, I recommend adding more comprehensive figures which should be contributions of the paper…
>
> A2. Thanks for the suggestions. We have moved the detailed results to the appendix, leaving only tables providing a concise and easy to understand summary for our results in the main paper.  We’ve added box plots and violin plots (Figure 1) to address this issue. We have additionally added plots indicating the spread of effect size for each of our 2 categories of diversity. Hoping this makes our results clearer and easier to understand.
>
>
> Q3. The results of Cohen's d can be caused from (1) meta-training dataset, and/or (2) meta-test dataset, in addition to learning algorithms, but I could not find which factor causes the results of Cohen's d. In other words, which diversity of meta-training datasets or meta-testing datasets makes the difference between the two methods?
>
> A3. We use the meta-training dataset to compute the Task2Vec diversity for each few-shot benchmark, as we believe that the data diversity of the meta-training should be the influencing factor that allows the meta-learner to meta-learn better. Since each few-shot task for meta-training and meta-testing is sampled from the same few-shot benchmark, the diversity of the test and training splits should be more or less the same.

---

> > ### Comment · Reviewer_RzB7 · 2023-12-03
> >
> > Thank you for your response. The added figure and tables are comprehensible compared to the ones before rebuttal. Unfortunately, even after rebuttal, the revised paper seems not yet of the quality for publication. For example, even though the main contributions should be experimental findings (since their proposed method is just a combination of existing methods), there are few results in the main text. I think the paper should contain more (comprehensible) results from several perspectives so that the reader can take some insights after reading the paper. Also, the whole page 6 is devoted to just listing experimental settings that are not essential to their claims, which deteriorates the quality of the paper. Thus, I still remain leaning towards the rejection side.

---

### Official Review · Reviewer_myHv · 2023-10-31

**Soundness:** 2 fair
**Presentation:** 1 poor
**Contribution:** 2 fair
**Rating:** 3
**Confidence:** 3

**Summary:**

* Paper examines the few-shot learning (FSL) paradigm, and studies the question of whether a pretrained (PT) model which then undergoes linear evaluation (finetuning of the final layer) is better than conducting meta-learning approaches such as MAML.
* This question has been studied before in different forms; the difference in this paper is that it standardizes the comparison using the same architecture for all methods, and trains models to convergence. A different statistical tool, Cohen's d, is used to assess performance differences. The usage of Cohen's d/effect sizes is because t-tests can result in very small p values/confidence intervals when the number of samples compared is large (which it is in this case).
* To study this question, the paper considers many different FSL datasets (21 totally) and compares the performance of PT models to MAML models.  The paper also evaluates the diversity score of datasets as a tool to help interpret the results, based on Task2vec (Miranda et al 2022).
* The key finding is that PT models are not always better than MAML-styled metalearned models (based on the effect size score for the comparisons). in particular, one class may perform better than the other as a function of the diversity of the dataset considered.

**Strengths:**

* The problem studied has value, especially with the field moving towards PT and finetuning/linear evaluation over metalearning approaches. Studying the problem in the context of the task diversity is particularly interesting, and yields some intuitive results -- we would possibly expect metalearning to perform better in the context of high diversity (over pretraining).

* discussion of why cohen's d was used is interesting -- paper has clearly put thought into choosing appropriate statistical measures to understand the results obtained.

* Diversity of datasets considered is valuable.

**Weaknesses:**

* Choice of MAML model: the original MAML model has been developed significantly in recent years. It would be fairer to that class of methods to compare to one of these many developments, given they have demonstrated (in general) better performance.

* A discussion of the drawbacks of effect size -- it is useful to understand where this may be inadequate (except the fact that one must choose a threshold level). Relatedly, the paper states that standard effect sizes are 0.2, 0.5, and 0.8, but it is unclear how the choice of 1% relates to these.

* Presentation of experimental setup: given this paper is primarily related to empirical benchmarking, a summary of the experimental setup in the main paper would help contextualise the investigation. This includes summary information about the datasets, abbreviations used (which reflect the results tables), training setup etc.

* Clarity in results exposition: when reporting the overall effect size, it is unclear how these are obtained -- my guess is an average over the cases where that hypothesis was determined correct. This is not mentioned (as far as I saw) -- it would help to do so.

* Clarity in results table: The tables could be better presented, for example by having the captions clearly specify that MAML5 refers to 5 adaptation steps, and having the same formatting for the heading and the main body (like all uppercase). Also, the detail on seeds in Table 3 is hard to understand -- what are seed1 and seed1 vs seed2 comparisons? The caption mentions 5 layer CNNs but the table refers to Resnet12 and Resnet50? Overall, the results are interesting but the presentation is very hard to follow.

* Training to convergence: Is this necessarily a good thing? If models experience some sort of overfitting, perhaps it makes sense to do early stopping?

* Minor: possibly missing citation in Section 5 when referring to feature reuse vs rapid learning

**Questions:**

See above; questions related to choice of MAML model, drawbacks of effect size, clarity in results tables.

---

> ### Author Response · Authors · 2023-11-22
> **Thank you for thoughtful review**
>
> We thank Reviewer myHv for the insightful comments. Below is our response:
>
> Q1. Choice of MAML model: the original MAML model has been developed significantly in recent years. It would be fairer to that class of methods to compare to one of these many developments, given they have demonstrated (in general) better performance.
>
> A1. We aim to counter a rising belief that meta-learning approaches are always worse than pre-training approaches. To do so, we pick the simplest meta-learning approach, MAML, and show that it can perform better than pre-training in high diversity settings. This is sufficient to counter the belief, since if MAML can outperform pre-training, so can the other meta-learning algorithms.
>
> Q2. A discussion of the drawbacks of effect size -- it is useful to understand where this may be inadequate (except the fact that one must choose a threshold level). Relatedly, the paper states that standard effect sizes are 0.2, 0.5, and 0.8, but it is unclear how the choice of 1% relates to these.
>
> A2. We acknowledge the reviewer’s point that it is important to understand the limitations of using any statistical test. One of the major limitations of using effect sizes is that they can be misleading if considered in isolation of other metrics such as meta-test accuracies. We present these values in the appendix, section A.2. We further note that, as with any other statistical test, we need to define arbitrary thresholds for the interpretation of effect sizes. While the common values are 0.2,0.5,0.8, we have used the 1% threshold as a decision rule based on it being common to accept papers due to 1% differences in machine learning. We have provided a detailed description of our choice of the 1% threshold in the appendix, section A.6, paragraph 3. Note that the choice of 1% doesn’t affect our primary results, in which we report average of raw effect size values.
>
> 	We highlight and answer the aforementioned issues in detail in Section 3.2 of our paper.
>
>
> Q3. Presentation of experimental setup: given this paper is primarily related to empirical benchmarking, a summary of the experimental setup in the main paper would help contextualise the investigation. This includes summary information about the datasets, abbreviations used (which reflect the results tables), training setup etc.
>
> A3. We have noted the hyperparameters and experimental setup in Section 4 in the revision (re-arranged from the appendix). While the information about datasets and abbreviations are still mentioned in the appendix, we have moved the detailed results section to the appendix, leaving only tables providing a concise and easy to understand summary for our results in the main paper. We have also added plots indicating the spread of effect size for each of our 2 categories of diversity. Hoping this makes our results clearer and easier to understand.
>
> Q4. Clarity in results exposition: when reporting the overall effect size, it is unclear how these are obtained -- my  guess is an average over the cases where that hypothesis was determined correct. This is not mentioned ... - it would help to do so.
>
> A4. Thanks for catching that. It is indeed the average over effect sizes that resulted in the same decision. We have added a brief explanation to the caption of table 1.
>
> Q5. Clarity in results table: The tables could be better presented, for example by having the captions clearly specify that MAML5 refers to 5 adaptation steps, and having the same formatting for the heading and the main body (like all uppercase). Also, the detail on seeds in Table 3 is hard to understand -- what are seed1 and seed1 vs seed2 comparisons? The caption mentions 5 layer CNNs but the table refers to Resnet12 and Resnet50? Overall, the results are interesting but the ...
>
> A5. Thank you for pointing these errors out. We have fixed all aforementioned errors in the revision. We have additionally moved the detailed results to the appendix, leaving only tables providing a concise and easy to understand summary for our results in the main paper. We have also added plots indicating the spread of effect size for each of our 2 categories of diversity. Hoping this makes our results clearer and easier to understand.
>
> Q6. Training to convergence: ... If models experience some sort of overfitting, perhaps it makes sense to do early stopping?
>
> A6. All datasets used in this paper have a sufficient number of examples per class for our models to be resistant to overfitting. We further back this claim by noting that we do not see poor test performance as should be expected for overfitting. Since our models do not overfit, it doesn’t make sense to implement early stopping, which would also otherwise hinder our ability to do a fair comparison between the two techniques. In addition, fitting the data perfectly is a standard procedure for ResNet & vision models on classification tasks.
>
> Q7. Minor: ... use vs rapid learning
>
> A7. We have added the citation in, thanks for the catch!

---

### Official Review · Reviewer_ZnA1 · 2023-11-02

**Soundness:** 2 fair
**Presentation:** 3 good
**Contribution:** 2 fair
**Rating:** 5
**Confidence:** 3

**Summary:**

This paper presents an extensive set of empirical results comparing pre-training and meta-learning across numerous model choices and datasets. The paper also proposes a new statistic test based on the classical concept of effect size, which allows for comparison across different domains.

The results show that on task space with low diversity, pre-training tends to perform better. Otherwise, meta-learning performs better based in the above statistical test.

Overall, the main contribution of this paper is an extensive empirical study that provides some interesting insights on when pre-training is better than MAML and vice versa.

**Strengths:**

The paper is sufficiently well-written.

It experimental results are also quite extensive, reporting several insightful observations.

There is also the development of a new statistics test, which is both novel and interesting.

**Weaknesses:**

Despite the above strength, I still have a few doubts regarding the proposed evaluation scheme:

1. It would be better if the authors can elaborate more on why p-value and confidence interval become zero, which in turn motivates the development of the new test

2. What is the main principle behind the new test? Specifically, if it rejects a hypothesis, what can we tell about its confidence in doing so? For example, using t-test, we are implicitly assuming the performance differences follows a student-t distribution and the t-statistic basically tells us if we can reject the null hypothesis at a certain confidence level? Do we have a similar principle under the new test?

3. For the task space with high diversity, I tend to think that the pre-training would not be effective will small architecture for obvious reason. Thus, the statistic test should only be applied to the reported performance on the best model architecture for pre-training (but across different test domains) -- right now, it seems the performance report on ResNet50 would favor pre-training; and the conclusion that meta-learning is better is mostly likely caused by the included performance on lower-complexity ResNet structure.

**Questions:**

Based on my concerns above, the following questions were raised:

1. Can the authors elaborate more on why p-value & confidence interval would become zero on larger batch size?
2. Please elaborate on the main principle behind a new test (i.e., its underlying assumption regarding the distribution of performance difference between two algorithms + what can we say about the confidence in rejecting the null hypothesis)
3. For high diversity setting, I think the statistic test should be based only on the best model architecture. Please consider revising this part of the evaluation protocol

---

> ### Author Response · Authors · 2023-11-22
> **Thank you for thoughtful review**
>
> We thank Reviewer ZnA1 for the insightful comments. Below is our response:
>
> Q1: Can the authors elaborate more on why p-value & confidence interval would become zero on larger batch size?
>
> A1: As the batch size increases, our sample size gets larger, and we tend to see that the confidence interval becomes zero (which can sometimes misleadingly suggest that the differences between two groups is significant - see https://www.nature.com/articles/s41598-021-00199-5). As addressed in Section 3.2, effect size is less affected by sample size which alleviates this drawback.
>
> Q2: Please elaborate on the main principle behind a new test (i.e., its underlying assumption regarding the distribution of performance difference between two algorithms + what can we say about the confidence in rejecting the null hypothesis)
>
> A2: Section 3.2 addresses why effect size is a good metric (compared to confidence intervals) to compare MAML and PT. In summary, effect size provides a way of quantifying the difference between the two algorithms - if the absolute value of the effect size is greater than the calculated standardized 1% threshold, we reject the null hypothesis. The effect size also quantifies the magnitude of the difference - the standard values are 0.2, 0.5, and 0.8 for a small, medium, and large effect size difference.
>
> Q3: For high diversity setting, I think the statistic test should be based only on the best model architecture. Please consider revising this part of the evaluation protocol
>
> A3: Our goal in this paper is to provide a fair comparison between pre-training and meta-learning, and to provide empirical evidence to the fact that meta-learning outperforms pre-training if the data diversity is large enough. We define a fair comparison as using the same model, same optimizer and training to convergence. We hence provide comparisons with different models and with different data diversities to conclude that high diversity and small architectures should lead to better meta-learning performance. With bigger architectures, the difference between meta-learning and pre-training is less pronounced due to the possibility of pre-trained models to meta-overfit.

---

> > ### Comment · Reviewer_ZnA1 · 2023-12-02
> > **Post-discussion thought**
> >
> > Thank you for the explanation on the principle behind the new test based on effect size. I understand the intuition but it still does not address the question on how confident we can be when we decline the null hypothesis.
> >
> > For example, if the absolute value of the effect size is greater than the calculated standardized 1% threshold, can we estimate the chance that the null hypothesis is correct? Without such provable justification, the test mechanism appears ad-hoc, and might not be robust across different application scenarios.
> >
> > In addition, the last point seems like a debatable point to me: if we take away the key advantage of pre-training in being able to leverage a large architecture over meta-learning which cannot easily accommodate high-complexity model, would the comparison still be meaningful -- e.g., why even consider pre-training with low-complexity model?
> >
> > Overall, I think the comparison here needs to be more extensive to draw out a comprehensive assessment regarding the advantage of meta-learning over pre-training. Otherwise, it is not clear if we are comparing the best of each solution family.
> >
> > --
> >
> > Taking into account the above, as well as the reviews from other reviewers, I will maintain my current rating of this paper.

---

### Meta-Review · Area_Chair_8z2u · 2023-12-06

**Metareview:**

This paper aims to introduce a fair and rigorous comparison of fine tuning versus metalearning for few shot problems. The authors accomplish this by controlling the architecture, optimizer, and convergence of models, while also using the Cohen’s D statistic to determine the significance of the difference.  The question investigated by this paper is significant and timely for the larger machine learning community, and I commend the authors for taking a statistically rigorous approach towards answering this question. Nevertheless, in the original submission, there were questions about the experimental setup, including (1) the use of MAML as the metalearning baseline (when more advanced methods exist), (2) the set of controls (e.g. does training to convergence potentially cause overfitting, causing the methods to be suboptimal), and (3) thoroughly explaining why effect size should be used to measure significance. Furthermore, there were concerns about clarity and presentation, as the results were not presented in a way that clearly communicated the main findings. Though the authors addressed these concerns in the rebuttal phase, the number of changes to the writing/experiments constitutes a “major revision” that I believe would benefit from an additional round of reviewing.

**Justification For Why Not Higher Score:**

The authors commented on the reviewers concerns in the rebuttal (see metareview) and have begun to address them in the revision. Nevertheless, this paper would benefit from an additional round of reviewing before it should be considered for a top-tier conference.

**Justification For Why Not Lower Score:**

N/A

---

### Decision · Program_Chairs · 2024-01-16

Reject